# Different brain systems support learning from received and avoided pain during human pain-avoidance learning

Marieke Jepma[1,2,3]*, Mathieu Roy[4,5], Kiran Ramlakhan[2,6], Monique van Velzen[7], Albert Dahan[7]

[1]Department of Psychology, University of Amsterdam, Amsterdam, Netherlands; [2]Department of Psychology, Leiden University, Leiden, Netherlands; [3]Leiden Institute for Brain and Cognition, Leiden, Netherlands; [4]Department of Psychology, McGill University, Montreal, Canada; [5]Alan Edwards Centre for Research on Pain, McGill University, Montreal, Canada; [6]Department of Research and Statistics, Municipality of Amsterdam, Amsterdam, Netherlands; [7]Department of Anesthesiology, Leiden University Medical Center, Leiden, Netherlands

**Abstract** Both unexpected pain and unexpected pain absence can drive avoidance learning, but whether they do so via shared or separate neural and neurochemical systems is largely unknown. To address this issue, we combined an instrumental pain-avoidance learning task with computational modeling, functional magnetic resonance imaging (fMRI), and pharmacological manipulations of the dopaminergic (100 mg levodopa) and opioidergic (50 mg naltrexone) systems ($N$ = 83). Computational modeling provided evidence that untreated participants learned more from received than avoided pain. Our dopamine and opioid manipulations negated this learning asymmetry by selectively increasing learning rates for avoided pain. Furthermore, our fMRI analyses revealed that pain prediction errors were encoded in subcortical and limbic brain regions, whereas no-pain prediction errors were encoded in frontal and parietal cortical regions. However, we found no effects of our pharmacological manipulations on the neural encoding of prediction errors. Together, our results suggest that human pain-avoidance learning is supported by separate threat- and safety-learning systems, and that dopamine and endogenous opioids specifically regulate learning from successfully avoided pain.

*For correspondence:
mariekejepma@gmail.com

Competing interest: The authors declare that no competing interests exist.

## Editor's evaluation

This manuscript is of particular interest to readers in the field of pain research. The identification of separate brain systems associated with learning from unexpected pain and learning from unexpected pain relief contributes to understanding of pain avoidance learning. The combination of behavioral and neuroimaging data and computational modeling analyses provide support for the central claims of the paper.

## Introduction

Learning to avoid actions that cause damage to our body is critical for health and survival. The experience of pain is an important teaching signal in this learning process, such as when a child learns to avoid touching a hot stove, or when a patient who underwent knee surgery learns to avoid bending his or her knee. However, the *absence* of otherwise expected pain can be an equally important teaching signal. When, for example, some weeks after surgery a patient realizes that bending his or her knee

is *not* painful anymore, this suggests that particular movements are safe again and no longer need to be avoided. Adaptive behavior in situations associated with pain thus requires an optimal balance between threat and safety learning when confronted with, respectively, the unexpected presence and absence of pain.

Previous studies have made considerable progress in our understanding of the neural basis of passive cue–pain association learning (*Ploghaus et al., 2000*; *Seymour et al., 2004*; *Seymour et al., 2005*) and—more recently—active pain-avoidance and -relief learning (*Roy et al., 2014*; *Eldar et al., 2016*; *Zhang et al., 2018*) in humans. A key aspect of these studies was the application of reinforcement-learning models to the analysis of neuroimaging data. According to reinforcement-learning theory, learning is driven by *prediction errors*, which signal the difference between the actual and expected outcomes of an action (*Sutton and Barto, 1998*). Thus, actions that result in the unexpected presence vs. absence of pain yield oppositely signed prediction errors which, respectively, increase and decrease the aversive value associated with that action. However, whether these opponent teaching signals drive learning via one underlying brain system, or via separate ones, is still largely unknown.

One possibility is that prediction errors elicited by the unexpected presence and absence of pain are encoded as opposite activity patterns in the same brain regions (i.e., one learning system). If this is the case, we may also expect that—at the neurochemical level—learning from these two outcomes is supported by the same neuromodulator(s). Furthermore, these two outcomes may then be equally effective in driving learning, such that they are associated with the same learning rate. Most previous studies, including our own, have assumed that this is the case. For example, in a previous functional magnetic resonance imaging (fMRI) study, we identified brain activity encoding *general* aversive prediction errors, signaling the degree to which *both* received- and avoided-pain outcomes are relatively worse (or less good) than expected (*Roy et al., 2014*). Another possibility, however, is that learning from received and successfully avoided pain are subserved by two separate brain systems. In this case, learning from these two outcomes may also be supported by different neuromodulatory systems, and associated with different learning rates. Although the notion of two systems is broadly consistent with theoretical accounts of avoidance learning, such as two-factor theory (*Mowrer, 1951*), there is not much empirical evidence for this idea, especially not in humans.

Regarding the role of neuromodulators, there is a wealth of evidence that reward prediction errors, which are thought to drive reward-pursuit behaviors, are signaled by phasic activity of midbrain dopamine neurons (*Schultz et al., 1997*) that project to the ventral striatum (*O'Doherty et al., 2003*, *Rutledge et al., 2010*). Specifically, unexpected rewards trigger a phasic increase (burst) in dopamine activity, while the unexpected absence of reward triggers a phasic decrease (dip) in dopamine activity. Whether the dopamine system also has a role in aversive learning is controversial, but several hypotheses have been proposed, mostly based on animal studies (*Palminteri and Pessiglione, 2017*). According to one prominent account, the same dopaminergic prediction-error response that reinforces actions associated with reward also reinforces actions associated with the omission of aversive outcomes (*Mowrer, 1956*; *Dinsmoor, 2001*; *Moutoussis et al., 2008*). This account thus predicts that learning from successfully avoided pain is supported by phasic dopamine bursts. However, animal studies have provided mixed evidence for this idea (*Josselyn et al., 2005*; *Oleson et al., 2012*; *Wietzikoski et al., 2012*; *Dombrowski et al., 2013*; *Fernando et al., 2013*; *Salinas-Hernández et al., 2018*; *Wenzel et al., 2018*; *Stelly et al., 2019*) and evidence in humans is scarce (*Raczka et al., 2011*). In addition, it has been proposed that negative punishment learning is mediated by phasic dips or pauses in dopamine activity, via an effect on D2 receptors (*Frank, 2005*; *Maia and Frank, 2011*). Evidence for his account in humans is largely based on studies using secondary reinforcers (e.g., monetary losses), and whether phasic dopamine dips also mediate learning from unexpected pain is yet unknown.

The endogenous opioid system is another interesting candidate neuromodulatory system for pain-avoidance learning. Pavlovian fear conditioning studies in animals have provided evidence for a causal role of opioidergic activity—specifically via μ-opioid receptors in the periaqueductal gray (PAG)—in both fear conditioning (*McNally and Cole, 2006*; *Cole and McNally, 2007*; *Cole and McNally, 2009*; *McNally et al., 2011*) and fear-extinction learning (*McNally and Westbrook, 2003*; *McNally et al., 2004*; *McNally et al., 2005*; *Kim and Richardson, 2009*; *Parsons et al., 2010*). These findings suggest that the endogenous μ-opioid system may mediate learning from received and/or avoided

pain, although this remains to be studied in instrumental learning tasks and in humans. In humans, endogenous opioids are well known to mediate pain relief (*Levine et al., 1978*; *Eippert et al., 2009*) as well as pleasure and liking responses (*Berridge and Kringelbach, 2015*; *Nummenmaa and Tuominen, 2018*), but it is currently unknown whether the roles of the opioid system in affective valuation also influence pain-related *learning*. One interesting possibility—that remains to be explored—is that endogenous opioids support pain-avoidance learning by enhancing the pleasant feeling of relief following successful pain avoidance (*Sirucek et al., 2021*).

In the present study, we aimed to dissociate learning from received and avoided pain in terms of behavior (learning rates), neural encoding of prediction errors, and the roles of dopamine and endogenous opioids. To this end, we combined an instrumental pain-avoidance learning task with computational modeling, fMRI, and pharmacological manipulations of the dopamine and opioid systems, in a randomized, double-blind, between-subject design. Specifically, participants completed the learning task under one of three drug conditions: 100 mg levodopa (a dopamine precursor), 50 mg naltrexone (an opioid antagonist, with highest affinity for the μ-opioid receptor), or placebo. PET studies in humans suggest that levodopa increases phasic dopamine bursts (*Floel et al., 2008*). Thus, if dopamine bursts support learning from successfully avoided pain, we expect levodopa to enhance learning rates and neural prediction-error signaling when pain is avoided. Additionally, although levodopa is thought to primarily increase phasic dopamine bursts (*Floel et al., 2008*; *Black et al., 2015*), it may increase tonic dopamine levels as well, which could in turn interfere with dopamine dips (*Breitenstein et al., 2006*; *Moustafa et al., 2013*; *Poletti and Bonuccelli, 2013*). Thus, if phasic dopamine dips support learning from unexpected pain, we expect levodopa to suppress learning rates and neural prediction-error signaling when pain is received. Naltrexone blocks the majority of μ-opioid receptors in the brain (*Lee et al., 1988*; *Preston and Bigelow, 1993*; *Schuh et al., 1999*; *Weerts et al., 2013*). Thus, if μ-opioid-receptor activity supports learning from received- and/or avoided-pain outcomes, we expect naltrexone to reduce learning rates and neural prediction-error signaling for the corresponding outcome(s).

## Results

Eighty-three healthy human participants completed a pain-avoidance learning task during fMRI, under one of three treatment conditions (levodopa, naltrexone, or placebo). On each of 144 trials of the pain-avoidance learning task, participants chose between two options (*Figure 1A*), each probabilistically associated with the delivery of painful heat (49 or 50°C, 1.9 s duration) to their left lower leg. Pain probabilities for each option were governed by two independently varying random walks (*Figure 1B*).

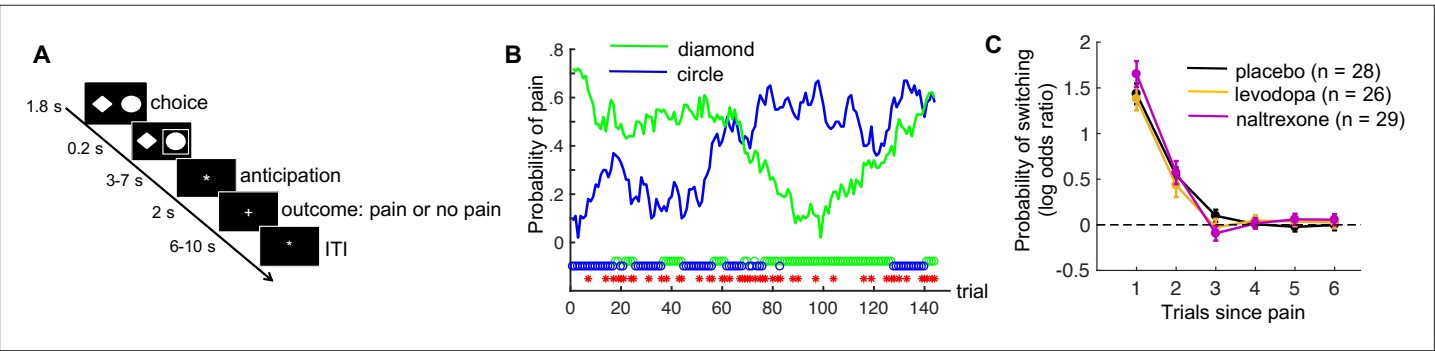

**Figure 1.** Pain-avoidance learning task. (**A**) Outline of one trial. (**B**) Example of pain probabilities and choice data for one participant. The green and blue lines show the trial-specific probabilities of receiving a painful heat stimulus when choosing each option. The green and blue circles below the graph indicate the participant's choices, and the red stars indicate trials on which pain was delivered. (**C**) Probability of switching per treatment group, as a function of pain 1–6 trials back. Error bars are standard errors.

The online version of this article includes the following figure supplement(s) for figure 1:

**Figure supplement 1.** Pre- and post-treatment ratings of alertness, calmness, and contentment in each treatment group.

**Figure supplement 2.** Pain ratings during a pain-rating task that preceded the pain-avoidance learning task, as a function of stimulus temperature and treatment group.

Thus, participants had to track the changing reinforcement values, and kept experiencing prediction errors throughout the task.

There were no treatment effects on subjective state (alertness, calmness, or contentment; *Figure 1—figure supplement 1*) or self-reported heat pain during a separate pain-rating task immediately preceding the avoidance-learning task (*Figure 1—figure supplement 2*).

Nine participants were excluded from the fMRI (but not the behavioral) analyses because of excessive head movement. Thus, the behavioral analyses included 83 participants (26–29 per treatment group), and the fMRI analyses included 74 participants (24–26 per treatment group).

## No drug effects on model-independent measures of task performance

On average, participants received pain on 57.1 of the 144 trials (standard deviation [SD] = 8.6). As expected, participants switched to the other choice option more frequently after receiving pain (46.6% of those trials, SD = 21.2) than after avoiding pain (5.4% of those trials, SD = 6.9; $t(82)$ = 18.6, $p < 0.001$, Cohen's $d$ = 2.0). The effect of previous pain outcomes on switching also decayed exponentially over time, in all treatment groups (*Figure 1C*; $p < 0.001$, Cohen's $d > 2.2$ for 1 trial back, $p < 0.002$, Cohen's $d > 0.7$ for 2 trials back, and $p > 0.047$, Cohen's $d < 0.4$ for 3–6 trials back, in all groups).

The three treatment groups did not differ in the number of received pain stimuli ($F(2,80)$ = 0.56, $p = 0.57$, $\eta^2 = 0.01$), frequency of switching following pain outcomes ($F(2,80)$ = 0.03, $p = 0.97$, $\eta^2 = 0.0007$), or frequency of switching following no-pain outcomes ($F(2,80)$ = 1.18, $p = 0.31$, $\eta^2 = 0.03$). Thus, our pharmacological manipulations did not affect basic measures of task performance.

Choice reaction times (RTs) were faster when participants stayed with the same choice option following a no-pain outcome (no pain/stay choices; mean = 729 ms) than when participants either stayed or switched following a pain outcome (mean = 778 and 795, respectively; no pain/stay vs. pain/stay: $t(82)$ = 5.0, $p < 0.001$, Cohen's $d$ = 0.54; no pain/stay vs. pain/switch: $t(82)$ = 6.9, $p < 0.001$, Cohen's $d$ = 0.76). Pain/stay and pain/switch RTs did not differ from each other ($t(82)$ = 1.7, $p = 0.10$, Cohen's $d$ = 0.18). The faster no pain/stay choices likely reflect that no-pain outcomes usually indicate that participants are on the right track, which makes staying with that option a relatively simple decision. The interpretation of pain outcomes is less straightforward, as these outcomes could either indicate a change in outcome probabilities—in which case a switch to the other option is warranted—or an occasional 'unlucky' outcome due to the probabilistic task nature. There was no treatment effect on RT for no pain/stay, pain/stay, or pain/switch trials ($F(2,80)$ = 0.35, $p = 0.70$, $\eta^2 = 0.01$; $F(2,80)$ = 0.68, $p = 0.51$, $\eta^2 = 0.02$; $F(2,80)$ = 0.93, $p = 0.40$, $\eta^2 = 0.02$, respectively).

## Computational modeling

To formalize and quantify the latent learning and decision processes thought to underlie participants' choice behavior, we applied two candidate reinforcement-learning models to the choice data, and compared their goodness-of-fit. Both models update the expected pain probability for the chosen option on each trial, in proportion to the prediction error (*Rescorla and Wagner, 1972*). The two models differ in that Model 1 uses a single learning rate, $\alpha$, for all outcomes, whereas Model 2 uses separate learning rates for received and avoided pain: $\alpha_{pain}$ and $\alpha_{no-pain}$, respectively (see Methods for model equations and parameter-estimation details). If Model 2 is better able to explain the choice data than Model 1, this could be taken as initial support for the idea that learning from received and avoided pain is subserved by different learning systems.

Both learning models were combined with a softmax decision function that translates expected pain probabilities into choice probabilities. Inverse-temperature parameter $\beta$ controls the degree of choice randomness, such that the likelihood that the model chooses the option with the lowest expected pain probability increases as $\beta$ increases.

We estimated the model parameters using a hierarchical Bayesian approach. This approach assumes that every participant has a different set of model parameters, which are drawn from group-level distributions. In this way, the information in the individual data is aggregated, while still respecting individual differences (*Gelman, 2014*). Each group-level distribution is in turn governed by a group-level mean and a group-level SD parameter. These group-level parameters were estimated separately for the placebo, levodopa, and naltrexone groups. As we are primarily interested in differences between treatment groups, our primary variables of interest are the parameters governing the means

of the group-level distributions, which we denote with overbars (e.g., $\bar{\alpha}_{pain}$ refers to the group-level mean of $\alpha_{pain}$).

## Model comparison: evidence for asymmetric learning from received and avoided pain

In all treatment groups, the model with separate learning rates for received and avoided pain (Model 2) outperformed the model with a single learning rate (Model 1), providing initial evidence for the presence of two learning systems. The Watanabe–Akaike information criterion (WAIC) for Model 1 vs. Model 2 was 3109 vs. 2959, 2958 vs. 2941, and 3164 vs. 3106 for the placebo, levodopa, and naltrexone groups, respectively.

## Parameter estimates: levodopa and naltrexone increase learning rates for avoided pain

We next examined the parameter estimates of the best-fitting model. *Figure 2A* shows the posterior distributions of the group-level mean parameters, per treatment group. The corresponding 95% highest density intervals (HDIs) are reported in *Figure 2—source data 1*. The 95% HDIs of each individual participant's learning-rate posteriors are shown in *Figure 2—figure supplement 1*.

In the placebo group, the posterior distribution of $\bar{\alpha}_{pain}$ (median = 0.72) was considerably higher than the posterior distribution of $\bar{\alpha}_{no-pain}$ (median = 0.32; $\bar{\alpha}_{pain} > \bar{\alpha}_{no-pain}$ for 99.6% of the Markov chain Monte Carlo [MCMC] samples), indicative of stronger expectation updating when pain was received than avoided. In contrast, in both drug groups, the posterior distributions of $\bar{\alpha}_{pain}$ and $\bar{\alpha}_{no-pain}$ were highly similar, due to a specific increase in $\bar{\alpha}_{no-pain}$ relative to the placebo group. In the levodopa group, the posterior medians of $\bar{\alpha}_{pain}$ and $\bar{\alpha}_{no-pain}$ were, respectively, 0.66 and 0.66 ($\bar{\alpha}_{pain} > \bar{\alpha}_{no-pain}$ for 50% of the MCMC samples). In the naltrexone group, the posterior medians of $\bar{\alpha}_{pain}$ and $\bar{\alpha}_{no-pain}$ were, respectively, 0.72 and 0.76 ($\bar{\alpha}_{pain} > \bar{\alpha}_{no-pain}$ for 38% of the MCMC samples). Note that the best-fitting model for both drug groups contained separate learning rates for received and avoided pain. Combined with the finding that the group-level mean learning-rate parameters for these two outcomes were highly similar, this suggests that some participants in each drug group learned more from received than avoided pain while others showed the opposite bias, but that there was no systematic learning asymmetry (at the individual level, $\alpha_{pain}$ was higher than $\alpha_{no-pain}$ for 50% of the levodopa and 41% of the naltrexone participants).

Thus, at the group level, both levodopa and naltrexone, as compared to placebo, increased learning rates for avoided pain, while not affecting learning from received pain (*Figure 2A*, left and middle panels). To test the significance of these group differences, we computed the difference between the posterior distributions of the group-level mean parameters for each drug group vs. the placebo group (*Figure 2B*). For $\bar{\alpha}_{no-pain}$ , 99.7% of the difference distribution for levodopa vs. placebo, and 99.9% of the difference distribution for naltrexone vs. placebo, lay above 0. In contrast, $\bar{\alpha}_{pain}$ did not differ between the drug and placebo groups: 34% and 49% of the difference distributions for levodopa vs. placebo and naltrexone vs. placebo, respectively, lay above 0. Thus, both drugs selectively increased learning rates for avoided pain.

The posterior distribution of inverse-temperature parameter $\bar{\beta}$ was higher for the placebo group (median = 8.8) than the levodopa and naltrexone group (median = 5.3 and 5.7, respectively), as well. Specifically, 98.8% of the $\bar{\beta}$ difference distribution for levodopa vs. placebo, and 98.1% for naltrexone vs. placebo, lay below 0, indicating that $\bar{\beta}$ was reliably lower for both drug groups compared to the placebo group. This suggests that participants in the two drug groups, as compared to the placebo group, were less prone to choose the option with the lowest expected pain probability (i.e., more stochastic choice behavior).

Together, the parameter estimates suggest that (1) untreated (placebo group) participants updated their expectations more rapidly following received than avoided pain, (2) levodopa and naltrexone negated this learning asymmetry by selectively increasing learning rates for avoided pain, and (3) levodopa and naltrexone also increased choice stochasticity, possibly reflecting a more exploratory or risky choice strategy.

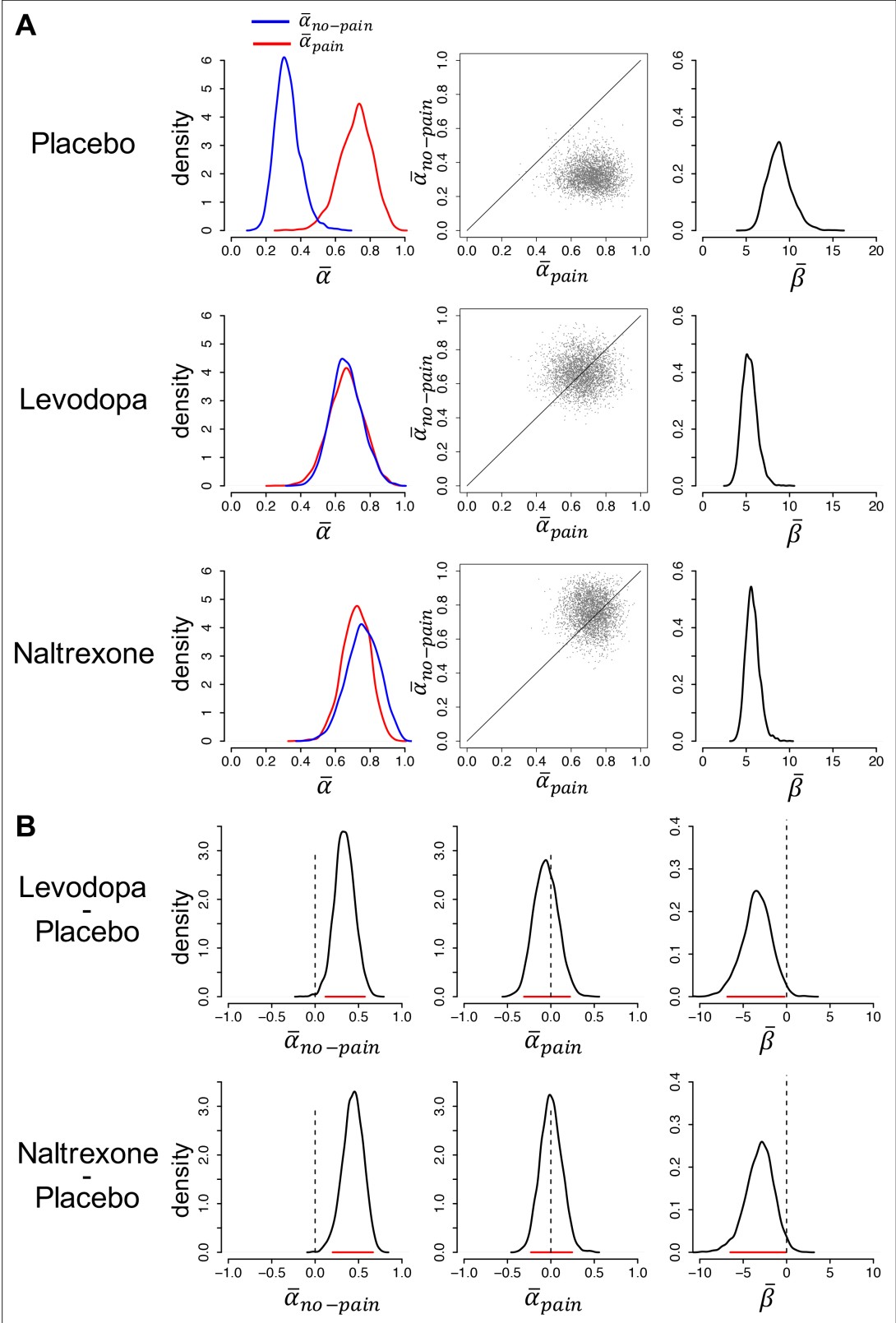

**Figure 2.** Model parameters. (**A**) Posterior distributions of the parameters' group-level means for each group (left and right panels). Parameters $\alpha_{no-pain}$ and $\alpha_{pain}$ are learning rates for avoided and received pain outcomes, respectively; parameter $\beta$ is the inverse-temperature parameter. The middle panels are joint density plots of $\overline{\alpha}_{pain}$ and $\overline{\alpha}_{no-pain}$ (dots are samples from the Markov chain Monte Carlo [MCMC]), showing that $\overline{\alpha}_{pain}$ is reliably greater than $\overline{\alpha}_{no-pain}$ in the placebo group only. (**B**) The difference between the posterior distributions for each drug group vs. the placebo

*Figure 2 continued on next page*

*Figure 2 continued*

group, showing that $\overline{\alpha}_{no-pain}$ is greater and $\alpha_{pain}$ is smaller in both drug groups compared to the placebo group. Red lines indicate 95% highest density intervals (HDIs).

The online version of this article includes the following source data and figure supplement(s) for figure 2:

**Source data 1.** The 95% highest density intervals (HDIs) of the posterior distributions of each participant's learning rate for pain ($\alpha_{pain}$) and no-pain ($\alpha_{no-pain}$) outcomes.

**Figure supplement 1.** The 95% highest density intervals (HDIs) of the posterior distributions of each participant's learning rate for pain ($\alpha_{pain}$) and no-pain ($\alpha_{no-pain}$) outcomes.

**Figure supplement 2.** Modeling results from an independent sample of untreated participants from a previous study ($N$ = 23), replicating the asymmetric learning rates ($\overline{\alpha}_{pain} > \overline{\alpha}_{no-pain}$) found in our placebo group.

## Replication of learning-rate asymmetry in an independent group of untreated participants

To examine the robustness of our finding of asymmetric learning rates in the placebo group, we applied our hierarchical Bayesian modeling approach to the choice data of 23 untreated participants from our previous pain-avoidance learning fMRI study (*Roy et al., 2014*), and tested whether the higher learning rate for received than avoided pain found in our placebo group was replicated in this previous dataset (*Figure 2—figure supplement 2*). In this previous dataset, $\overline{\alpha}_{pain}$ (median = 0.62) was indeed higher than $\overline{\alpha}_{no-pain}$ (median = 0.44), resembling the placebo-group results from our current study, although the learning-rate asymmetry was somewhat smaller in the previous dataset ($\overline{\alpha}_{pain} > \overline{\alpha}_{no-pain}$ for 90% of the MCMC samples). The posterior median of $\overline{\beta}$ in the previous dataset was 7.1 (95% HDI = 4.7–9.9). The finding of a higher learning rate for pain than no-pain outcomes in this independent dataset corroborates the idea that people normally (in the absence of a pharmacological manipulation) update their expectations more rapidly following received than avoided pain.

## Data simulation and parameter recovery

The fact that our pharmacological manipulations increased $\overline{\alpha}_{no-pain}$ (negating the learning asymmetry found in the placebo group) and reduced $\overline{\beta}$ (increasing choice stochasticity) seems at odds with the absence of drug effects on model-independent performance measures. A possible explanation for these results is that the observed performance measures (e.g., switch/stay behavior) reflect a combination of several underlying variables—including the learning rates for received and avoided pain and the degree of choice stochasticity—and that the drug effects on learning rate and choice stochasticity canceled each other out. Specifically, symmetric learning rates for received and avoided pain (as found in the drug groups) likely resulted in more accurate pain-probability estimates than asymmetric learning rates (as found in the placebo group). This beneficial effect of a symmetric learning process in the drug groups, however, may have been counteracted by the detrimental effect of a more stochastic choice process, such that the combination of these two computational effects resulted in no net performance difference between the placebo and drug groups. However, since $\alpha$ and $\beta$ are generally negatively correlated to one another (*Cools et al., 2011*), it is also possible that the observed tradeoff between $\overline{\alpha}_{no-pain}$ and $\overline{\beta}$ reflected an artifact of the parameter-optimization procedure rather than 'true' effects of the pharmacological manipulations.

To rule out the possibility of an artifactual consequence of the parameter-optimization procedure, we simulated two sets of choice data on our task using the parameter values found in the placebo and drug groups, respectively, and performed parameter-recovery analyses (Appendix 1). In sum, our modeling procedure was able to correctly recover the simulated parameters, suggesting that it is unlikely that the parameters observed in the different pharmacological-manipulation conditions are artifacts of our parameter-optimization procedure. Moreover, the simulated datasets also produced similar model-independent performance measures (number of received pain stimuli, switch frequencies) in the drug and placebo groups, despite the fact that the simulated choices were produced by different sets of parameters. These findings suggest that levodopa and naltrexone really had two computational effects—an increased learning rate for avoided pain, and an increased degree of choice stochasticity—which combination yielded no significant effects on model-independent performance measures.

## fMRI analyses

To address the question whether learning from received and avoided pain is supported by two separate brain systems, we sought to identify brain activation encoding *outcome-specific* prediction-error signals. We focused on the first second of the outcome period as this is when prediction errors are triggered. We modeled drug effects using two second-level regressors (levodopa vs. placebo and naltrexone vs. placebo). fMRI results are thresholded at *q* < 0.05, false discovery rate (FDR) corrected for multiple comparisons across the whole brain (gray matter masked). Unthresholded *t* maps can be found on https://neurovault.org/collections/RIVRRMAK/. When referring to subdivisions of the anterior cingulate cortex (ACC), we use the terms rostral and dorsal ACC, as is common in neuroimaging studies (*Bush et al., 2000*). Note that these two regions have also been referred to as ACC and midcingulate cortex, respectively (*Palomero-Gallagher et al., 2009*).

We also performed an axiomatic analysis to identify brain activation encoding *general* aversive prediction errors (i.e., activation encoding the degree to which both pain and no-pain outcomes are relatively worse, or less good, than expected). In our previous study, this analysis revealed a general aversive prediction-error signal in the PAG (*Roy et al., 2014*). Here, we examined whether we could replicate this finding. As this analysis is not directly linked to the research questions addressed in the present study, we report it in Appendix 2.

## Outcome-specific prediction-error signals

Regions encoding the unexpectedness or surprise evoked by received pain should respond stronger to pain outcomes when pain was less expected (i.e., a negative correlation with expected pain probability on pain trials). In contrast, regions encoding the surprise evoked by avoided pain should respond stronger to no-pain outcomes when pain was more expected (i.e., a positive correlation with expected pain probability on no-pain trials; *Figure 3A*, left plots). To identify regions that show a stronger surprise response for received than avoided pain, we thus specified the following contrast: negative correlation with expected pain probability on pain trials > positive correlation with expected pain probability on no-pain trials. This contrast revealed activation in the vmPFC and rostral ACC, posterior cingulate cortex, insula extending into the left parahippocampal gyrus, cerebellum, and a brainstem region encompassing part of the PAG (*Figure 3A*, yellow regions). We also found extensive clusters that showed the opposite effect—that is positive correlation with expected pain probability on no-pain trials > negative correlation with expected pain probability on pain trials—in sensorimotor cortex, parietal and occipital cortex, and the bilateral frontal poles (*Figure 3A*, blue regions), suggesting that these regions show a stronger surprise response for avoided than received pain.

We also sought to identify activation encoding the surprise elicited by both received and avoided pain, that is, activation encoding absolute prediction errors. To this end, we specified a second contrast that tested for a negative correlation with expected pain probability on pain trials *and* a positive correlation with expected pain probability on no-pain trials (*Figure 3B*). This contrast revealed extensive activation clusters in the dorsal ACC extending into the supplementary motor cortex, insula, sensorimotor cortex, thalamus, part of the brainstem and cerebellum, suggesting that these regions encoded absolute prediction error (*Figure 3B*, yellow regions). In addition, a few smaller clusters in left sensorimotor cortex, right dlPFC, and left frontopolar cortex showed the opposite effect, suggesting that these regions encoded the overall *expectedness* of outcomes (*Figure 3B*, blue regions).

Finally, note that a caveat of the first contrast reported above (*Figure 3A*) is that it also identifies activation that is stronger when the expected pain probability is lower (i.e., activation encoding expected safety), regardless of the outcome. This likely explains the vmPFC activation, which has been found to represent positive affective value in many domains (*Roy et al., 2012*; *Bartra et al., 2013*; *Rich and Wallis, 2016*). The second contrast (for absolute prediction errors; *Figure 3B*), on the other hand, will *not* detect activation encoding expected safety regardless of the outcome, as the correlation with expected pain probability is specified in opposite directions for pain and no-pain outcomes. Thus, we reasoned that regions identified by both of the contrasts reported in *Figure 3A, B* encode outcome-specific prediction errors (*Figure 3A*) unconfounded by outcome-nonspecific expected pain probability (*Figure 3B*). Therefore, we next examined the conjunction of these two contrasts. Specifically, we masked the activation identified by the first contrast (separately for the positive and negative activation) by the positive activation identified by the second contrast. Positive activation for both contrasts was found in a set of mostly subcortical and limbic regions, including a large cluster

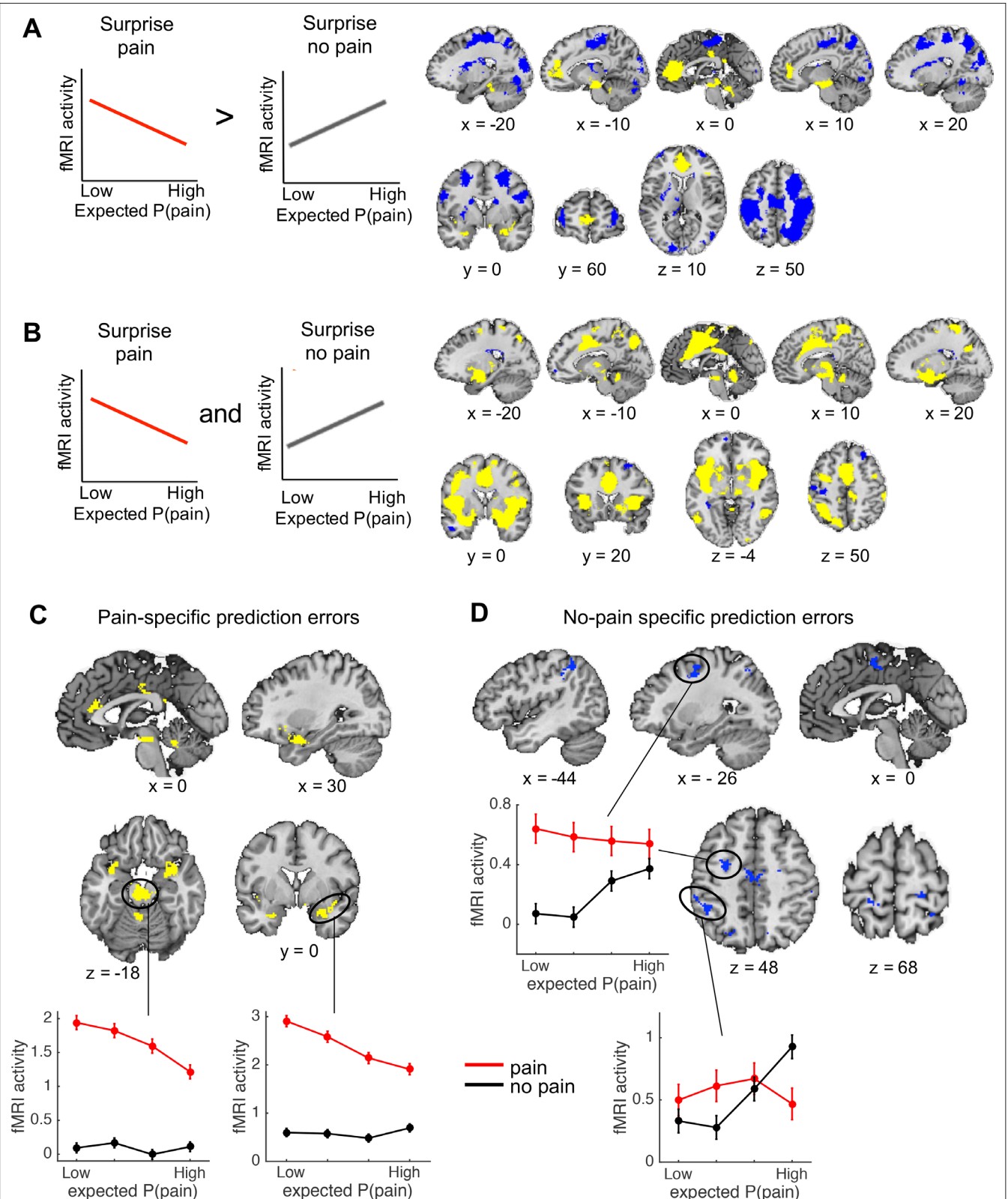

**Figure 3.** Outcome-specific prediction-error signals (*N* = 74). (**A**) Activation tracking surprise more for received than avoided pain (yellow) and vice versa (blue). Note that this includes activation that tracks expected pain probability across both outcomes. Expected P(pain) = expected pain probability. (**B**) Activation tracking surprise for both received and avoided pain (i.e., absolute prediction error). Activation maps in A and B are thresholded at $q < 0.05$, false discovery rate (FDR) corrected for multiple comparisons across the whole brain. (**C**) Regions encoding surprise more for received than avoided

*Figure 3 continued on next page*

*Figure 3 continued*

pain, which cannot be explained by a general sensitivity to expected pain probability. These regions showed positive activation for both the first (**A**) and second (**B**) contrast, each thresholded at $q < 0.05$, FDR corrected. (**D**) Regions encoding surprise more for avoided than received pain, which cannot be explained by a general sensitivity to expected pain probability. These regions showed negative activation for the first, and positive activation for the second contrast, each thresholded at $q < 0.05$, FDR corrected. The line plots show the mean activity extracted from the brainstem and right amygdala (**C**) and left dlPFC and parietal (**D**) clusters per quartile of expected pain probability, illustrating the encoding of outcome-specific prediction errors in these regions. Error bars are standard errors.

in the brainstem (not covering the PAG), bilateral insula extending into the amygdala, rostral ACC, posterior cingulate cortex, bilateral supramarginal gyrus, and cerebellum (*Figure 3C*). These regions thus encoded surprise more for received than avoided pain (pain-specific prediction errors), which could not be explained by a general sensitivity to expected pain probability. Negative activation for the first contrast and positive activation for the second contrast, on the other hand, was found in several cortical regions, including the supplementary motor cortex, left parietal cortex (supramarginal gyrus), bilateral somatosensory cortex (postcentral gyrus), and left dlPFC (middle and superior frontal gyrus) (*Figure 3D*). These regions thus encoded surprise more for avoided than received pain (no-pain-specific prediction errors), which could not be explained by a general sensitivity to expected pain probability. Together, these results provide evidence that prediction errors evoked by pain and no-pain outcomes are encoded in largely distinct brain regions.

## No drug effects on surprise-related brain activation

Because levodopa and naltrexone specifically increased learning rates for no-pain outcomes, we expected these drugs to increase prediction-error-related brain activation for no-pain outcomes as well. However, we found no differences between the levodopa and placebo group, or between the naltrexone and placebo group, for any of the contrasts reported above (whole-brain FDR corrected). Drug effects were virtually absent at lower, uncorrected, thresholds as well (see unthresholded *t* maps on https://neurovault.org/collections/RIVRRMAK/).

## Discussion

Our results provide novel evidence that unexpectedly received and avoided pain—signaling threat and safety, respectively—drive human pain-avoidance learning via different learning systems. First, computational modeling suggested that participants' choices were best explained by a model with separate learning rates for received and avoided pain, and that untreated participants learned more from received than avoided pain. Second, levodopa and naltrexone selectively increased learning rates for avoided pain, suggesting a role for the dopamine and endogenous-opioid systems in safety, but not threat, learning. Third, our fMRI analyses revealed that different brain circuits encode prediction errors elicited by received and avoided pain, providing evidence for two dissociable learning systems at the neural level as well. Somewhat surprisingly, however, we found no drug effects on fMRI activity for any of the contrasts we examined. We discuss each of these findings below.

### Learning rates for received vs. successfully avoided pain

The higher learning rate for pain than no-pain outcomes in the placebo group suggests that people normally (in the absence of a pharmacological manipulation) update their expectations more following received than avoided pain. A similar learning-rate asymmetry was found in a recent aversive reversal-learning study (*Wise et al., 2019*). Interestingly, however, reward-learning studies using secondary reinforcers (e.g., monetary gains and losses) have provided evidence for the *opposite* asymmetry: higher learning rates for favorable than unfavorable outcomes. This has been attributed to an optimistic learning bias (*Sharot and Garrett, 2016*; *Lefebvre et al., 2017*) and a tendency to learn preferentially from information that confirms one's current action (*Palminteri et al., 2017*). The opposite learning bias in pain-avoidance learning tasks may be due to the intrinsically aversive nature of pain, arguably rendering unexpected pain a more salient teaching signal than unexpected pain absence. Relatedly, the experience of pain may trigger a reflexive tendency to change one's course of action (*Huys et al., 2012*), expressed in elevated learning rates for pain outcomes. Thus, higher learning rates for received than avoided pain may reflect a Pavlovian influence on choice which operates in

parallel to the instrumental learning system. Alternatively, the seemingly opposite learning asymmetries in reward-learning and pain-avoidance learning tasks may also reflect a cognitive process related to the framing of the task. That is, participants who are instructed to maximize reward vs. minimize pain may pay most attention to—and hence learn most from—reward vs. pain outcomes, respectively.

The presence and direction of learning asymmetries may also depend on the specific task demands. For example, a previous fMRI study that used a more complex pain-avoidance learning task (pain probabilities of three different options were learned in parallel, in an indirect manner), and included a risk-taking component, found no systematic difference in learning rates for received and avoided pain (*Eldar et al., 2016*). Interestingly, in contrast to our present and previous (*Roy et al., 2014*) fMRI findings, the PAG in that study *positively* encoded expected pain probability on no-pain trials (one of the axioms for appetitive prediction errors), and did not encode expected pain probability on pain trials. These findings suggest that the task used in that previous study (*Eldar et al., 2016*) evoked different learning processes than our simpler task. Indeed, learning rates in that previous study were much lower than those found in our task as well. Further examination of the degree to which avoidance-learning processes and their neural implementation generalize across different learning tasks is an important objective for future work (*Yarkoni, 2020*).

## Effects of levodopa and naltrexone on learning parameters

Levodopa and naltrexone selectively increased learning rates for successfully avoided pain, consistent with a role for dopamine and endogenous opioids in safety learning. Previous studies have associated levodopa-induced increases in phasic dopamine activity with enhanced learning from (secondary) rewards (*Frank et al., 2004*; *Pessiglione et al., 2006*). Combined with these previous findings, our levodopa results suggest that phasic dopamine activity may signal the degree to which outcomes are 'better than expected' across both reward and punishment domains. We are aware of one previous study that provided correlational evidence for a role of dopamine in human safety learning in a Pavlovian fear-conditioning task (*Raczka et al., 2011*). That study found that individual differences in fear-extinction learning rates were associated with genetic variation in the dopamine transporter gene, which presumably affects phasic striatal dopamine release. Our levodopa results are consistent with this result, and provide the first *causal* evidence for a role of dopamine in human safety learning in an instrumental learning task. Levodopa did not affect learning rates for received pain; hence our results do not support the idea that dopamine supports learning from aversive outcomes. Instead, our results suggest a selective role for the human dopamine system in learning from successfully avoided pain.

Regarding the endogenous opioid system, we expected that if pain-avoidance learning relies on µ-opioid activity, naltrexone—which blocks this activity—would *suppress* learning from received and/or avoided pain. However, we found the opposite effect for avoided-pain outcomes: like levodopa, naltrexone increased learning rates for avoided pain. This finding counterintuitively suggests that µ-opioid activity normally suppresses learning from avoided pain and that naltrexone countered this effect, which seems to contradict findings that µ-opioid-receptor antagonists impair fear-extinction learning in rats (*McNally and Westbrook, 2003*; *McNally et al., 2004*; *McNally et al., 2005*; *Kim and Richardson, 2009*; *Parsons et al., 2010*). Obviously, these animal studies differed from our study in several ways—such as the nature of the learning task (Pavlovian vs. instrumental), outcome measure (freezing behavior vs. learning-rate estimates), opioid-receptor antagonist (naloxone vs. naltrexone), and drug administration (injection into the PAG vs. oral administration)—each of which may explain the apparently contradictory results. Thus, our results show that untreated participants learn more rapidly from received than avoided pain (replicated in our previous study) and that both levodopa and naltrexone negate this learning asymmetry, but the neurobiological mechanisms underlying the naltrexone effect remain to be elucidated. One informative approach for future studies would be to directly compare effects of opioid-receptor agonists and antagonists on pain-avoidance learning parameters.

The levodopa and naltrexone groups also showed a higher degree of choice stochasticity than the placebo group, suggesting that participants in both drug groups were less prone to choose the option with the lowest expected pain probability. Importantly, our parameter-recovery analysis indicated that the drug effects on learning rate for avoided pain and choice stochasticity could be independently and correctly retrieved. The increased choice stochasticity in the levodopa group may reflect a positive

association between dopamine activity and risk preference (*Voon et al., 2006*; *Gallagher et al., 2007*; *St Onge and Floresco, 2009*; *Chew et al., 2019*) or exploration (*Beeler et al., 2010*; *Kayser et al., 2015*; *Gershman and Tzovaras, 2018*). It is also broadly consistent with recent evidence that levodopa reduces the impact of valence on information seeking (*Vellani et al., 2020*). We are not aware of previous studies that associated the endogenous opioid system with choice stochasticity, risk taking, or exploration. It is possible that the similar effects of levodopa and naltrexone on choice stochasticity were due to common general side effects of both drugs on participants' attention or motivation, which disrupted the decision-making process. However, we believe this is unlikely because (1) the drugs did not affect subjective state (alertness, calmness, or contentment) and (2) general side effects cannot easily explain the specific increase in learning rates for avoided pain.

Interestingly, the drug effects on learning rate for avoided pain and choice stochasticity were not accompanied by drug effects on basic performance measures (number of received pain outcomes, pain-switch or avoid-switch behavior). In a similar vein, previous studies have reported effects of pharmacological manipulations and dopaminergic genotype on model parameters despite a lack of significant effects on behavioral measures (*Raczka et al., 2011*; *Chakroun et al., 2020*), illustrating the added value of computational models. Our simulation results suggested that the drug effects on learning rate and choice stochasticity in our study canceled each other out. Specifically, symmetric learning rates for received and avoided pain (found in the drug groups) result in more accurate pain-probability estimates than asymmetric learning rates (found in the placebo group). This beneficial effect of a symmetric learning process in the drug groups was, however, counteracted by the detrimental effect of a more stochastic choice process, resulting in no net performance difference between the placebo and drug groups.

## Separate brain circuits support learning from received and avoided pain

Our fMRI results provided evidence for two dissociable learning systems at the brain level as well. Pain-specific prediction errors were predominantly encoded in subcortical (brainstem, cerebellum) and limbic (insula, amygdala, rostral ACC) regions that are typically associated with emotional and affective processes, including fear conditioning (*Phillips and LeDoux, 1992*) and affective responses to errors (*Bush et al., 2000*). In contrast, no-pain-specific prediction errors were encoded in frontal and parietal cortical areas typically associated with higher-order cognitive processing (*Ptak et al., 2017*).

Prediction errors for no-pain outcomes were not represented in the ventral striatum, which is typically found for reward prediction errors (*O'Doherty et al., 2003*, *Rutledge et al., 2010*). Instead, prediction errors for no-pain outcomes were associated with frontoparietal activity. This activity is unlikely to reflect a reward signal, but possibly reflected increased attention on trials in which pain was expected but not received. That is, the unexpected absence of pain may have prompted participants to carefully monitor the thermode's temperature in order to verify whether pain was really avoided or still to come, as reflected in increased frontoparietal activity. The absence of a striatal 'reward-like' prediction-error response for no-pain outcomes suggests that, in terms of neural processing, learning from avoided pain is not comparable to learning from rewards. However, the lack of a detectable reward-like prediction-error response may also be related to our task design. Specifically, pain outcomes in our task involved a change in sensory input (a rise in temperature) whereas no-pain outcomes did not (maintenance of the baseline temperature), which may have caused a more prominent neural prediction-error response for the pain outcomes. One way to examine this issue would be to use a task in which choices result in either an increase or a decrease in painful stimulus intensity from a tonic pain level, such that aversive and appetitive-like outcomes are associated with similar changes in sensory input (*Seymour et al., 2005*). Such a task would examine pain-relief rather than pain-avoidance learning. The current task design, however, more closely resembles the situation of a patient recovering from injury or surgery, who is pain free as long as he or she is resting but expects that physical activity may result in pain. It is an interesting speculation that, in such situations, the stronger subcortical and limbic ('emotional') responses for pain than for no-pain prediction errors, as found in our study, may foster a behavioral state that favors inactivity and rest, which could promote tissue healing and recovery.

## No effects of levodopa and naltrexone on fMRI activation

Unexpectedly, we found no effects of levodopa or naltrexone on any of our fMRI prediction-error contrasts. This may indicate that the dopamine and opioid systems are not involved in pain-avoidance learning, although this is inconsistent with the drug effects on learning rates for avoided pain. It is also possible that our pharmacological manipulations *did* affect prediction-error-related dopamine and/or opioid activity, but that we did not have enough statistical power to detect these effects due to our moderate sample size and between-subject design (24–26 participants per treatment group). We opted for a between-subject design because we expected people's previous (possibly drug-related) experiences with our task to affect their motivation and behavior during subsequent sessions. For example, drug-induced impairments in avoidance learning could result in a perceived lack of control over the task outcomes—or learned helplessness (*Seligman and Maier, 1967*)—which could carry over to subsequent sessions. However, a disadvantage of our choice for a between-subject design is that it comes with less statistical power than a within-subject design.

The absence of drug effects on fMRI activation at the group level may also be explained by individual differences in drug responses. Several studies have shown that effects of dopaminergic drugs on cognitive function depend on an individual's baseline level of performance or dopamine activity reviewed in *Cools and D'Esposito, 2011*, *Frank and Fossella, 2011*, consistent with the idea that the relationship between dopamine activity and neurocognitive function follows an inverted U-shape function (*Cools and Robbins, 2004*). Thus, levodopa may have affected pain-avoidance learning and related brain function in opposite directions in different participants, depending on their baseline level of dopamine activity. Such baseline-dependent drug effects could have been studied in within-subject designs (in which each participant completes both a placebo and a drug session), ideally combined with a measure of participants' baseline dopamine/opioid level. In addition, future studies could use more than one drug dose to sample the putative inverted U-shape function.

Finally, it is important to note that drug effects on prediction-error-related dopamine and/or opioid activity may not always produce corresponding changes in the BOLD signal (*Knutson and Gibbs, 2007*; *Brocka et al., 2018*). This last possibility is clearly discouraging for pharmacological fMRI studies, but is conceivable given the absence of consistent levodopa effects on reward prediction-error signals in previous fMRI studies: One study reported an increased reward prediction-error signal in an levodopa group compared to a haloperidol group (although neither of these groups differed from a placebo group) (*Pessiglione et al., 2006*), but two recent studies using within-subject designs found no levodopa effects on neural reward prediction-error signals (*Kroemer et al., 2019*; *Chakroun et al., 2020*). To better understand the degree to which drug-induced changes in human neuromodulatory (e.g., dopamine) activity are detectable using fMRI, future research could directly compare drug effects on local changes in neuromodulator activity—for example, using molecular imaging procedures such as PET—with effects of these same drugs on the BOLD signal.

## Limitations and directions for future research

As mentioned above, a lack of statistical power due to our moderate sample size and between-subject design may have prevented the detection of drug effects in our fMRI analyses. Another limitation of our study is the absence of a pharmacological manipulation check, for example via blood samples and/or autonomic or behavioral measures that are known to be influenced by dopamine or opioid activity, such as eye blink rate for dopamine (*Jongkees and Colzato, 2016*). Our drug doses were based on previous pharmacological studies with between-subject designs which reported significant drug effects (*Pessiglione et al., 2006*; *Eippert et al., 2008*; *Guitart-Masip et al., 2012*; *Oei et al., 2012*; *Beierholm et al., 2013*; *Bunzeck et al., 2014*; *Guitart-Masip et al., 2014*; *Wittmann and D'Esposito, 2015*), and the drug effects on learning parameters in the present study suggests that our pharmacological interventions were effective as well. However, given the absence of a manipulation check, it is unknown whether our null findings are due to a true lack of dopaminergic/opioidergic regulation of pain-avoidance learning, an ineffectiveness of our pharmacological manipulations (e.g., because the doses were too low), or a lack of statistical power. Thus, the absence of drug effects on our fMRI results and model-independent performance measures should be taken with caution.

A limitation of our experimental design is that we did not acquire pain ratings during the pain-avoidance learning task. Therefore, we cannot exclude the possibility that our pharmacological manipulations affected participants' sensitivity to the pain outcomes during this task. This is especially

relevant for naltrexone as the μ-opioid system is known to be involved in nociceptive processing, and blockade of this system had been associated with pain inhibition. However, we believe that it is unlikely that naltrexone affected participants' pain sensitivity because (1) we found no effects of naltrexone on pain ratings immediately prior to the learning task, and (2) previous studies suggest that opioid antagonists rarely affect pain perception in experimental pain paradigms (*Grevert and Goldstein, 1977*; *Eippert et al., 2008*; *Werner et al., 2015*; *Sirucek et al., 2021*).

Our fMRI data suffered from signal dropout in inferior parts of the prefrontal cortex (including the orbitofrontal cortex), which is a common problem in fMRI studies using echo-planar imaging (*Ojemann et al., 1997*; *Deichmann et al., 2003*). Therefore, our results are agnostic with respect to the contribution of ventral prefrontal areas to pain-avoidance learning, and their potential modulation by levodopa or naltrexone. In addition, we did not collect physiological data during fMRI scanning; hence could not remove potential artifacts related to cardiac and respiratory processes. Correcting for physiological noise may have resulted in a higher signal-to-noise ratio and hence increase the significance of the results, especially in brainstem regions (*Linnman et al., 2012*; *Brooks et al., 2013*). Importantly, it is unlikely that physiological noise was synchronized with our experimental design (e.g., the application of pain outcomes) and thereby produced 'false positive' fMRI results because (1) frequencies of cardiac and respiration cycles are much higher than the frequency of our pain outcomes, and (2) we used jittered interstimulus intervals. Furthermore, even if physiological changes in respiratory and cardiac cycle did correlate with the timing of our pain outcomes, this would be unlikely to systematically affect our prediction-error signals as these were based on parametric-modulator regressors that were orthogonal to the main outcome-onset regressors.

Finally, regarding the role of neuromodulators, we focused on dopamine and endogenous opioids, but other neuromodulators are almost certainly involved in pain-avoidance learning as well. In particular, future work may focus on the serotonergic system, which has traditionally been associated with aversive processing, behavioral inhibition, and 'fight or flight' responses in rodents (*Deakin, 1983*; *Soubrié, 1986*, *Deakin and Graeff, 1991*). More recent pharmacological and genetic studies, which mostly used tasks with secondary reinforcers, have provided evidence for a role of the human serotonergic system in various aspects of aversive learning (*Chamberlain et al., 2006*; *Cools et al., 2008*; *Crockett et al., 2012*; *Hindi Attar et al., 2012*; *Robinson et al., 2012*; *den Ouden et al., 2013*). Based on these findings, it has been proposed that serotonin acts as an opponent to dopamine by mediating behavioral inhibition in response to punishment (*Daw et al., 2002*; *Dayan and Huys, 2009*; *Boureau and Dayan, 2011*; *Cools et al., 2011*). When generalizing these findings to the domain of pain-avoidance learning, we may expect a specific role for the serotonergic system learning from received pain (threat learning). However, other studies have found effects of serotonin manipulations on both reward and punishment learning (*Palminteri et al., 2012*; *Guitart-Masip et al., 2014*), and a study in which participants simultaneously learned probabilities of monetary rewards and painful shocks suggested that serotonin selectively modulates reward processing (*Seymour et al., 2012*). Taken together, previous work suggests that the serotonergic system has highly intricate and multifaceted roles in affective learning, likely due to its large number of receptor types, widespread projections, and interactions with other neuromodulators (*Dayan and Huys, 2009*). Future studies are required to further clarify the functional role(s) of serotonin in pain-avoidance learning.

## Conclusion

In sum, our results suggest that received and avoided pain drive human pain-avoidance learning via two different learning systems, in terms of both learning rates and the neural encoding of prediction errors. In addition, our computational-modeling results provide evidence for a causal role of the dopamine and endogenous opioid systems in learning from avoided, but not received, pain. Future studies are needed to elucidate the neural mechanisms via which our dopamine and opioid manipulations affected learning rates for avoided pain, and to reveal the potential role of other neuromodulators in pain-avoidance learning.

## Materials and methods

### Participants

Ninety-one healthy students (18–26 years old; 71% female; all right handed) took part in the study. Participants reported no history of psychiatric, neurological, or pain disorders, and no current pain. Participants were instructed to abstain from using alcohol or recreational drugs 24 hr prior to testing, and to not eat or drink (except for water) 2 hr prior to testing. The study was approved by the medical ethics committee of the Leiden University Medical Center (P15.116), and all participants provided written informed consent. Participants received a fixed amount of €60 plus a variable bonus of maximally €5 related to their performance on an additional task.

Six participants were excluded from all analyses because of thermode failure, and two additional participants because of poor task performance (see 'Pain-avoidance learning task'). In addition, nine participants were excluded from the fMRI, but not the behavioral, analyses because of excessive head movement (>3 mm in any direction). Thus, the final behavioral analyses included 83 participants (placebo group: $N$ = 28, mean age = 21.2, 74% female; levodopa group: $N$ = 26, mean age = 20.8, 73% female; naltrexone group: $N$ = 29, mean age = 20.8, 68% female). The final fMRI analyses included 74 participants (placebo group: $N$ = 24, mean age = 21.3, 71% female; levodopa group: $N$ = 24, mean age = 20.9, 71% female; naltrexone group: $N$ = 26, mean age = 20.8, 69% female). Sample size was based on previous studies that detected effects of dopamine (levodopa) or opioid (naloxone) manipulations on behavioral and/or fMRI measures using between-subject designs (*Pessiglione et al., 2006*; *Eippert et al., 2008*; *Guitart-Masip et al., 2012*; *Oei et al., 2012*; *Beierholm et al., 2013*; *Bunzeck et al., 2014*; *Guitart-Masip et al., 2014*; *Wittmann and D'Esposito, 2015*). These previous studies used sample sizes ranging from 13 to 30 participants per treatment group. We aimed at a sample size at the higher end of that range (25–30 participants per treatment group).

### General procedure

Two to fourteen days prior to the fMRI session, we assessed participants' eligibility using a general health questionnaire and an fMRI safety screening form. During this screening session, participants also practiced the pain-avoidance learning task.

Each eligible participant took part in one fMRI session. On the day of the fMRI session, participants received a single oral dose of either 100 mg levodopa, 50 mg naltrexone, or placebo, according to a double-blind, randomized, between-subject design. Levodopa was combined with 25 mg carbidopa—a decarboxylase inhibitor that does not cross the blood–brain barrier—to inhibit the conversion of levodopa to dopamine in the periphery. Approximately 30 min after drug administration, participants were positioned in the MRI scanner, after which we acquired a high-resolution structural scan. Approximately 53 min after drug administration, participants completed a 5-min pain-rating task during which they received a series of (unavoidable) heat stimuli of varying temperatures and rated their experienced pain following each stimulus (*Figure 1—figure supplement 2*). This task was included to test for drug effects on subjective pain responses, and to select a painful yet tolerable temperature for each participant in the pain-avoidance learning task. Sixty minutes after drug administration, roughly corresponding with peak plasma concentrations of levodopa and naltrexone, participants started the pain-avoidance learning task (described below), which lasted approximately 45 min. Following the pain-avoidance learning task, participants performed an 8-min probabilistic reward-learning task (not reported here).

We measured participants' subjective state at the beginning (before drug administration) and end (2 hr after drug administration) of the test session, by means of visual analog scales measuring alertness, calmness, and contentment (*Bond and Lader, 1974 Bond and Lader, 1974*; *Figure 1—figure supplement 1*). Both subjective state measures were collected outside the scanner.

### Pain-avoidance learning task

This instrumental pain-avoidance learning task contained 144 trials, divided in 4 runs of 36 trials. On each trial, participants made a choice between two options (a diamond and a circle). Choosing each option was associated with a specific probability of receiving a painful heat stimulation. The probabilities of receiving pain when choosing each option drifted across trials according to two independent random walks (*Figure 1A*). We used three different pairs of random walks (each pair crossed at least

once); each pair was administered to approximately one-third of the participants in each treatment group.

Each trial started with the presentation of the two choice options, randomly displayed at the left and right side of the screen for 1800 ms (*Figure 1B*). During this period, participants had to select one option by pressing a left or right button of the response unit, using their right index or middle finger, respectively. If participants did not respond in time (1.2% of trials), the computer randomly selected an option for them. The chosen option was highlighted for 200 ms, followed by an anticipation period of 3, 5, or 7 s during which a white asterisk (*) was presented in the center of the screen. Then the outcome—a painful heat stimulus applied to participant's leg for 1.9 s (see 'Thermal stimulation') or no stimulus—was presented. Outcome onset was accompanied by a change of the central asterisk to a colored plus sign (+). The plus sign was red or green during the first 200 ms of each pain and no-pain outcome, respectively, after which it turned white for the remainder of the outcome period. The color change was meant to prevent outcome uncertainty during the initial phase of the outcome period. Each trial ended with an intertrial interval of 6, 8, or 10 s during which an asterisk was presented. Except for the outcome probabilities, participants were fully informed about the task structure and procedure.

During the fMRI session, one participant switched choices more frequently following the absence of pain than following pain, and one other participant did deliberately not make a choice on 17% of the trials, due to the use of irrelevant strategies. We excluded these participants from further analysis.

## Thermal stimulation

Heat stimuli (ramp rate = 40°C/s; 1 s at target temperature) were applied to the inner side of participants' left lower leg using a Contact Heat-Evoked Potential Stimulator (CHEPS; 27 mm diameter Peltier thermode; Medoc Ltd., Israel). After the initial pain-rating task, 16% of the participants (four in the placebo, four in the levodopa, and seven in the naltrexone group) indicated that they would not tolerate repeated stimulation at the highest temperature they had received so far (50°C). For those participants, we used a temperature of 49°C in the pain-avoidance learning task. For the remaining participants, we used a temperature of 50°C. Between stimulations the thermode maintained a baseline temperature of 32°C. The total duration of each stimulation was 1850 ms (425 ms ramp-up and ramp-down periods, 1 s at target temperature) for 49°C stimuli and 1900 ms (450 ms ramp-up and ramp-down periods, 1 s at target temperature) for 50°C stimuli. After each scan run, we moved the thermode to a new site on the participant's leg. To reduce the impact of potential site-specific habituation (*Jepma et al., 2014*), we administered one initial heat stimulus before starting the first trial on a new skin site.

## Behavioral analyses

We tested whether the total number of received pain stimuli, the proportion of pain trials followed by a switch to the other choice option, and the proportion of no-pain trials followed by a switch to the other choice option differed between the three treatment groups, using one-way analyses of variance. In addition, for each treatment group, we used logistic regression to analyze the probability of switching choices as a function of outcome (pain vs. no-pain) over the six previous trials.

## Computational modeling

We fitted two reinforcement-learning (*Q* learning) models to participants' choice data: one model with a single learning rate (Model 1), and one model with separate learning rates for pain and no-pain outcomes (Model 2). On each trial $t$, both models update the expected probability of pain for the selected stimulus $s$, $Q_s$, in response to the experienced outcome $O$ (pain or no pain), according to:

$$Q_{s,t+1} = Q_{s,t} + \alpha \left( O_t - Q_{s,t} \right)$$

where $\left( O_t - Q_{s,t} \right)$ reflects the prediction error on each trial. We coded $O$ as −1 and 0 for pain and no-pain outcomes, respectively, and initialized the $Q$ values of both stimuli to −0.5. The value of the unchosen stimulus is not updated. Learning-rate parameter $\alpha$ controls *how much* $Q$ values are updated in response to each new outcome, such that higher values of $\alpha$ result in faster updating. The only difference between our two models is that Model 1 uses a single learning rate $\alpha$ for all outcomes,

whereas Model 2 uses separate learning rates for pain and no-pain outcomes: $\alpha_{pain}$ and $\alpha_{no-pain}$, respectively.

Both models were combined with a softmax decision function, which computes the probability of choosing stimulus $s$ on trial $t$ $(P_{s,t})$ as:

$$P_{s,t} = \frac{e^{Q_{s,t}*\beta}}{\sum\limits_{s'=1}^{2} e^{Q_{s',t}*\beta}}$$

Inverse-temperature parameter $\beta$ controls the sensitivity of choice probabilities to differences in $Q$ values. If $\beta$ is 0, both stimuli are equally likely to be chosen, irrespective of their expected pain probabilities. As $\beta$ increases, the probability that the model chooses the stimulus with the lower expected pain probability increases. Thus, Model 1 has two free parameters ($\alpha$ and $\beta$) and Model 2 has three free parameters ($\alpha_{pain}$, $\alpha_{no-pain}$, and $\beta$).

## Parameter estimation

We estimated the model parameters with a hierarchical Bayesian approach, using the hBayesDM package (*Ahn et al., 2017*). The hierarchical Bayesian approach assumes that every participant has a different set of model parameters, which are drawn from group-level prior distributions (*Gelman, 2014*). The parameters governing the group-level prior distributions (hyperparameters) are also assigned prior distributions (hyperpriors). We estimated separate group-level parameters for the placebo, levodopa, and naltrexone groups. To test for treatment effects, we compared the posterior distributions of the group-level means (i.e., the hyperparameters governing the means of the group-level distributions) for each drug group vs. the placebo group.

### Prior distributions

We used weakly informative priors, as implemented in the hBayesDM package. The group-level distributions for all individual parameters were assumed to be normal distributions. As hyperpriors for the mean and SD hyperparameters of these group-level distributions we used, respectively, a normal distribution with mean 0 and SD 1 and a positive half-Cauchy distribution with location 0 and scale 5 (*Ahn et al., 2017*). We transformed the parameters' unconstrained values to a [0,1] range using the inverse probit transformation (*Wetzels et al., 2010*; *Ahn et al., 2014*). In addition, we transformed $\beta$ to a [0,20] range by multiplying its inverse-probit transformed values by 20 (*Ahn et al., 2017*).

### MCMC sampling

The hBayesDM package performs hierarchical Bayesian parameter estimation with an MCMC sampling algorithm called Hamiltonian Monte Carlo (HMC) implemented in Stan and its R instantiation RStan. We ran 4 independent MCMC chains with different starting values, which each generated 5000 posterior samples. We discarded the first 1000 iterations of each chain as burn-in. In addition, we only used every fifth iteration to remove autocorrelation. Consequently, we obtained 3200 representative samples (800 per chain) per parameter per model fit. All chains converged (Rhat values <1.1).

## Model comparison

We compared the fit of our two models using the WAIC (*Watanabe, 2010*). The WAIC provides an estimate of a model's out-of-sample predictive accuracy, adjusted for the number of free parameters, in a fully Bayesian way. Lower WAIC values indicate better out-of-sample predictive accuracy. WAIC is reported on the deviance scale (*Gelman, 2014*) hence a difference in WAIC value of 2–6 is considered positive evidence, a difference of 6–10 strong evidence, and a difference >10 very strong evidence for one model over another (*Kass and Raftery, 1995*).

## Computation of expected pain probability for use in fMRI analyses

We computed trial-specific expected pain probabilities—to be used as parametric modulator regressors in the fMRI analyses—by applying our winning model to each participant's sequence of choices and outcomes. In line with previous studies (*Pine et al., 2010*; *Voon et al., 2010*; *Seymour et al., 2012*; *Hauser et al., 2015*; *Kroemer et al., 2019*), we used the exact same model to generate fMRI

regressors for all participants by instantiating the model with the mean learning-rate parameters (the individual-level posterior medians, averaged across all participants).

## fMRI acquisition and analysis

### Imaging acquisition

We acquired fMRI data on a 3T Philips Achieva MRI system (Best, The Netherlands) at the Leiden University Medical Center, using a standard whole-head coil. Stimulus presentation and data acquisition were controlled using E-Prime software (Psychology Software Tools). Visual stimuli were presented via a mirror attached to the head coil, and participants responded with their right hand via an MRI-compatible response unit. The pain-avoidance learning task was divided across four scan runs. Each run lasted 581 s (264 TRs), after discarding the first 5 TRs which served as dummy scans. Functional images were acquired with a T2*-weighted whole-brain echo-planar imaging sequence (TR = 2.2 s; TE = 30 ms; flip angle = 80°, 38 transverse slices oriented parallel to the anterior commissure-posterior commissure line, voxel size = 2.75 × 2.75 × 2.75 mm + 10% interslice gap). In addition, we acquired a high-resolution T1-weighted scan (TR = 9.8 ms; TE = 4.6 ms, flip angle = 8°, 140 slices, 1.17 × 1.17 × 1.2 mm, FOV = 224 × 177 × 168), at the beginning of the scan session.

## Preprocessing

Prior to preprocessing, global outlier time points (i.e. 'spikes' in the BOLD signal) were identified by computing both the mean and the SD (across voxels) of values for each image for all slices. Mahalanobis distances for the matrix of slice-wise mean and SD values (concatenated) × functional volumes (time) were computed, and any values with a significant $\chi^2$ value (corrected for multiple comparisons based on the more stringent of either FDR or Bonferroni methods) were considered outliers. On average 3.7% of images were outliers (SD = 1.9). The output of this procedure was later used as a covariate of noninterest in the first-level models.

Functional images were slice-acquisition-timing and motion corrected using SPM8 (Wellcome Trust Centre for Neuroimaging, London, UK). Structural T1-weighted images were coregistered to the first functional image for each subject using an iterative procedure of automated registration using mutual information coregistration in SPM8 and manual adjustment of the automated algorithm's starting point until the automated procedure provided satisfactory alignment. Structural images were normalized to MNI space using SPM8, interpolated to 2 × 2 × 2 mm voxels, and smoothed using a 6-mm full-width at half maximum Gaussian kernel.

## Analysis of outcome-specific prediction-error signals

For the first-level analysis, we create a general linear model for each participant, concatenated over the four pain-avoidance learning blocks, in SPM8. We modeled periods of decision time (cue onset until response, mean response time = 758 ms), outcome anticipation (3–7 s), onsets of pain outcomes (1 s), and onsets of no-pain outcomes (1 s), using boxcar regressors convolved with the canonical hemodynamic response function. As in our previous study (*Roy et al., 2014*), we only modeled the first second of the outcome periods as this is when prediction errors are triggered. We added the model-derived expected pain probability as a parametric modulator on the outcome-anticipation and outcome-onset regressors. To control for potential effects of outcome-anticipation duration, we also included anticipation duration as a first parametric modulator on the outcome-onset regressors (using serial orthogonalization, such that any shared variance between expected pain probability and anticipation duration is assigned to the anticipation-duration effect). Other regressors of noninterest (nuisance variables) were (1) 'dummy' regressors coding for each run (intercept for each but the last run); (2) linear drift across time within each run; (3) the 6 estimated head movement parameters (*x*, *y*, *z*, roll, pitch, and yaw), their mean-zeroed squares, their derivatives, and squared derivatives for each run (total 24 columns per run); (4) indicator vectors for outlier time points identified based on their multivariate distance from the other images in the sample (see above); (5) indicator vectors for the first two images in each run. Low-frequency noise was removed by employing a high-pass filter of 180 s.

To examine outcome-specific prediction-error signals, we created two contrast maps. First, to identify activation that tracks surprise more for received than avoided pain (*Figure 3A*), we used the following contrast: 'negative correlation with expected pain probability at pain onset' > 'positive correlation with expected pain probability at no-pain onset'. Second, to identify activation

tracking absolute prediction error (*Figure 3B*), we used the following contrast: 'negative correlation with expected pain probability at pain onset' > 'negative correlation with expected pain probability at no-pain onset'. Note that this contrast is identical to: 'positive correlation with expected pain probability at no-pain onset' > 'positive correlation with expected pain probability at pain onset'. It is also identical to a contrast with weights [1 1] for the 'negative correlation with expected pain probability at pain onset' and positive correlation with expected pain probability at no-pain onset' regressors.

We performed a second-level analysis on each of these two contrasts using robust regression, including two second-level regressors coding for levodopa vs. placebo (weights [−1 1 0] for the treatment groups [P L N]) and naltrexone vs. placebo (weights [−1 0 1] for the treatment groups [P L N]). Maps were thresholded at FDR $q < 0.05$ corrected for multiple comparisons across the whole brain.

Finally, to identify regions encoding outcome-specific prediction errors which cannot be explained by a general sensitivity to expected pain probability (also see 'Results') we examined the conjunction between the two second-level contrast maps described above, each thresholded at FDR $q < 0.05$ (*Figure 3C*).

## Acknowledgements

We thank Stephanie Bauduin, Laila Franke, Nikki Nibbering, Iliana Samara, and Iris Spruit for help with data collection.

## Additional information

### Funding

| Funder | Grant reference number | Author |
|---|---|---|
| Nederlandse Organisatie voor Wetenschappelijk Onderzoek | Veni | Marieke Jepma |

The funders had no role in study design, data collection, and interpretation, or the decision to submit the work for publication.

### Author contributions

Marieke Jepma, Conceptualization, Data curation, Formal analysis, Funding acquisition, Investigation, Methodology, Project administration, Supervision, Validation, Visualization, Writing - original draft; Mathieu Roy, Conceptualization, Writing – review and editing; Kiran Ramlakhan, Investigation, Writing – review and editing; Monique van Velzen, Albert Dahan, Resources, Writing – review and editing

### Author ORCIDs

Marieke Jepma (iD) http://orcid.org/0000-0002-7903-0135
Mathieu Roy (iD) http://orcid.org/0000-0002-3335-445X

### Ethics

All participants provided written informed consent study. The study was approved by the medical ethics committee of the Leiden University Medical Center (P15.116).

### Decision letter and Author response

Decision letter https://doi.org/10.7554/eLife.74149.sa1
Author response https://doi.org/10.7554/eLife.74149.sa2

## Additional files

### Supplementary files
• Transparent reporting form

## Data availability

Single-trial behavioral data and model code are available from the OSF database: https://osf.io/rqc6g/. Unthresholded t maps for all reported fMRI contrasts are available on Neurovault: https://identifiers.org/neurovault.collection:6016.

The following datasets were generated:

| Author(s) | Year | Dataset title | Dataset URL | Database and Identifier |
|---|---|---|---|---|
| Jepma M | 2021 | Two systems for pain-avoidance learning | https://osf.io/rqc6g/ | Open Science Framework, rqc6g |
| Jepma M | 2021 | Two brain systems for pain-avoidance learning | https://identifiers.org/neurovault.collection:6016 | NeuroVault, neurovault.collection:6016 |

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

# Appendix 1

## Parameter-recovery analysis

### Procedure

We simulated 100 datasets, each consisting of choices from 27 synthetic participants on the pain-avoidance learning task. The number of trials and pain probabilities was the same as for the real participants (each of our three random-walk pairs was assigned to 9 synthetic participants in each dataset). We set the means of the group-level distributions for $\alpha_{pain}$, $\alpha_{no-pain}$, and $\beta$ to the posterior medians obtained from our fits to either the placebo group or the levodopa/naltrexone group, in 50 simulations each (as the posterior medians for the two drug groups were highly similar, we used their average values). We set the SDs of the group-level distributions to 0.1 for $\alpha_{pain}$ and $\alpha_{no-pain}$, and to 1 for $\beta$, in all simulations. In each simulation, the individual-level parameters were randomly drawn from their group-level distributions.

We fitted our winning model (Model 2) to each of the simulated datasets using the same hierarchical Bayesian procedure as used for the real data, and compared the recovered to the simulated (true) group-level mean parameters. In addition, we computed pairwise difference distributions (placebo − drug) for the recovered group-level mean parameters, resulting in 50 difference distributions per parameter. For each parameter, we then computed the proportion of difference distributions whose 95% HDI did not include zero (i.e., the probability that a group difference was detected in the simulated datasets).

Finally, we tested if our model-free performance measures (number of times pain was received and frequencies of switching after received and avoided pain) differed between the datasets that were simulated with our placebo and drug parameters.

## Results

The recovered posterior medians of $\bar{\alpha}_{pain}$, $\bar{\alpha}_{no-pain}$, and $\bar{\beta}$ were clustered around their simulated (true) values, for both the placebo and drug simulations (*Appendix 1—figure 1*). Recovered values of $\bar{\alpha}_{no-pain}$ and $\bar{\beta}$ were negatively correlated ($r = -0.80$, p < 0.001 and $r = -0.51$, p < 0.001 for the placebo and drug simulations, respectively), reflecting the typical tradeoff between learning rate and inverse temperature (*Cools et al., 2011*). In addition, there was a positive correlation between the recovered values of $\bar{\alpha}_{pain}$ and $\bar{\beta}$ for the placebo simulations ($r = 0.38$, $P = 0.006$)—which was driven by the data point with the highest recovered $\bar{\beta}$ value (the correlation was not significant when excluding this data point)—but not for the drug simulations ($r = -0.02$, p = 0.9). The recovered values of $\bar{\alpha}_{pain}$ and $\bar{\alpha}_{no-pain}$ were uncorrelated (p's > 0.75 for both the placebo and drug simulations).

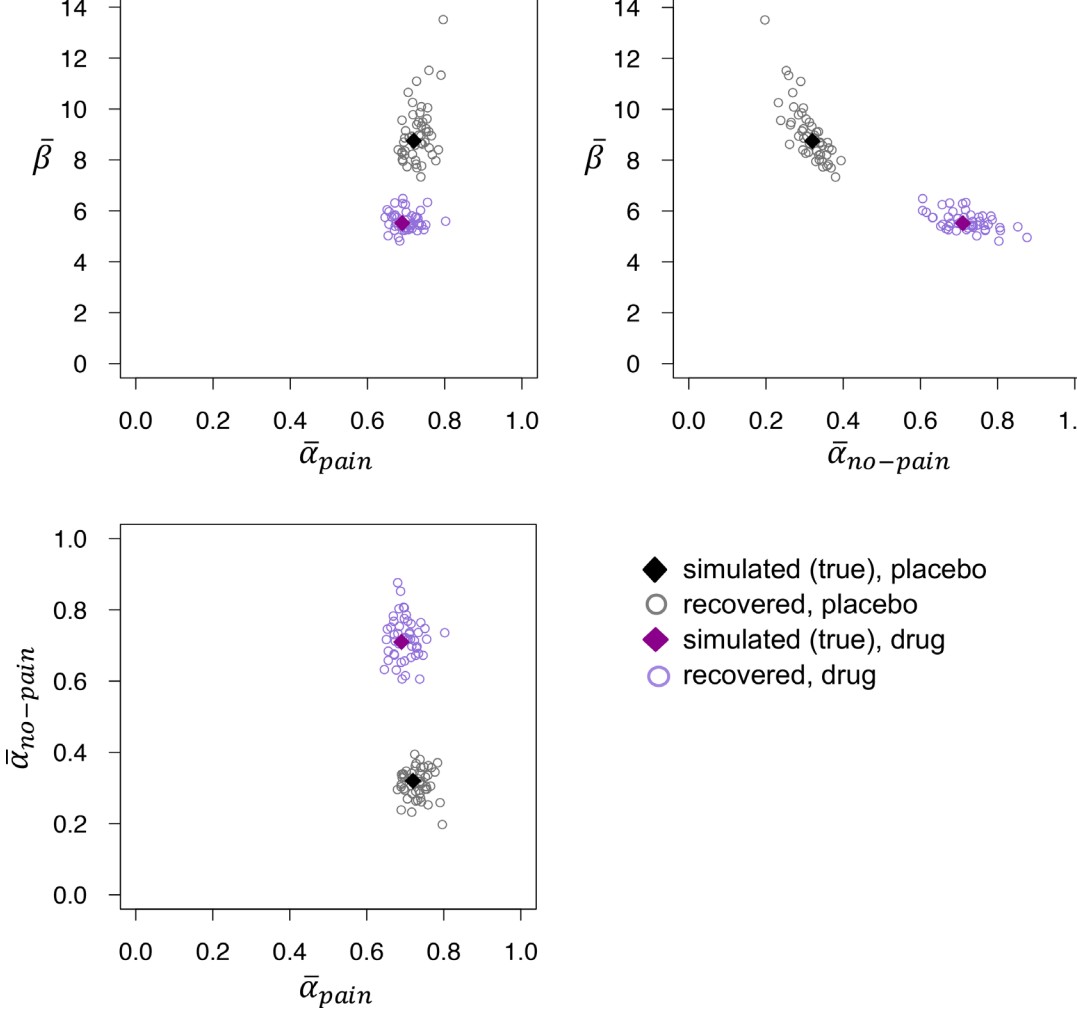

**Appendix 1—figure 1.** Recovered posterior medians of $\bar{\alpha}_{pain}$, $\bar{\alpha}_{no-pain}$, and $\bar{\beta}$ for fits to datasets that were simulated using the posterior medians from the placebo (black) and drug (purple) groups. The recovered values of $\bar{\alpha}_{pain}$ do not differ between the two groups, but recovered $\bar{\alpha}_{no-pain}$ is reliably higher and recovered $\bar{\beta}$ reliably lower in the drug group, mirroring the parameter estimates obtained from fits to the empirical data.

Despite the correlation between $\bar{\alpha}_{no-pain}$ and $\bar{\beta}$, recovered $\bar{\alpha}_{no-pain}$ was significantly higher for fits to datasets simulated with the drug, as compared to placebo, parameters (none of the 50 difference distributions' 95% HDI included zero). In addition, recovered $\bar{\beta}$ was significantly higher for fits to datasets simulated with the placebo parameters (none of the 50 difference distributions' 95% HDI included zero). The recovered values of $\bar{\alpha}_{pain}$ did not differ reliably between the drug and placebo simulations (80% of the difference distributions' 95% HDI included zero). These findings imply that our modeling procedure can accurately dissociate the two patterns of parameter values we found in our placebo and drug groups.

Our model-independent performance measures—number of pain stimuli received, and frequencies of switching after received and avoided pain—did not differ between the datasets that were simulated with the placebo and drug parameters (*Appendix 1—table 1*), mirroring the absence of treatment effects on these measures in the empirical data. Thus, our modeling procedure can distinguish between the two parameter patterns found in our placebo and drug groups, even though these produce the same model-independent performance measures.

**Appendix 1—table 1.** Model-free performance measures in simulated datasets that were generated using the group-level mean parameters from our placebo and drug groups.

| | Placebo simulations | Drug simulations | Average p value |
|---|---|---|---|
| Number of pain stimuli | 54.9 | 55.1 | 0.63 |
| Switching after pain | 45.0% | 44.1% | 0.44 |
| Switching after no pain | 4.8% | 3.7% | 0.41 |

Notes: Performance measures are averaged across all synthetic participants from each of 50 simulated datasets (27 synthetic participants per dataset). We performed 50 *t*-tests—each comparing the scores from the synthetic participants from one placebo vs. one drug simulation—and report their average p values.

In sum, the parameter-recovery results suggest that the differences between the placebo and drug groups in both $\alpha_{no-pain}$ and $\beta$ did not merely reflect a tradeoff between these two parameters or an artifact of the parameter-optimization procedure. Instead, these results suggest that the drugs had two computational effects—an increased learning rate for avoided pain, and an increased level of decision noise—whose combination caused no significant effects on basic, model-independent performance measures.

## Appendix 2

## General (outcome-nonspecific) aversive and appetitive prediction-error signals

Prediction-error-related activation is often examined by regressing fMRI activity at outcome onset on model-derived prediction errors. However, as prediction errors are defined as the outcome minus the expected outcome, a problem with this approach is that the resulting brain activity may predominantly track the outcome (in our task: pain vs. no pain) or the expected outcome (in our task: the expected pain probability), which are intrinsically correlated with the prediction error.

To address this issue, and identify brain activity that truly integrates actual and expected outcomes into a prediction-error signal, a set of conditions has recently been specified (*Rutledge et al., 2010*; *Roy et al., 2014*). These conditions, or axioms, for general aversive prediction-error signals in our task are: (1) activation at outcome onset should be higher for received than avoided pain, unless pain is fully expected; (2) when pain is received, activation should be higher when pain was less expected (i.e., negative correlation with expected pain probability); and (3) when pain is avoided, activation should also be higher when pain was less expected, that is, when avoidance was more expected (*Appendix 2—figure 1A*, left panels). To identify regions encoding a general aversive prediction-error signal, we tested for activation that fulfilled each of these three axioms, using a whole-brain conjunction analysis. In addition, to search for regions encoding the opposite (i.e., appetitive-like) prediction-error signal, we also tested for activation that fulfilled each of the *reverse* axioms. Note that we did not detect activation encoding appetitive-like prediction errors in our previous study (*Roy et al., 2014*).

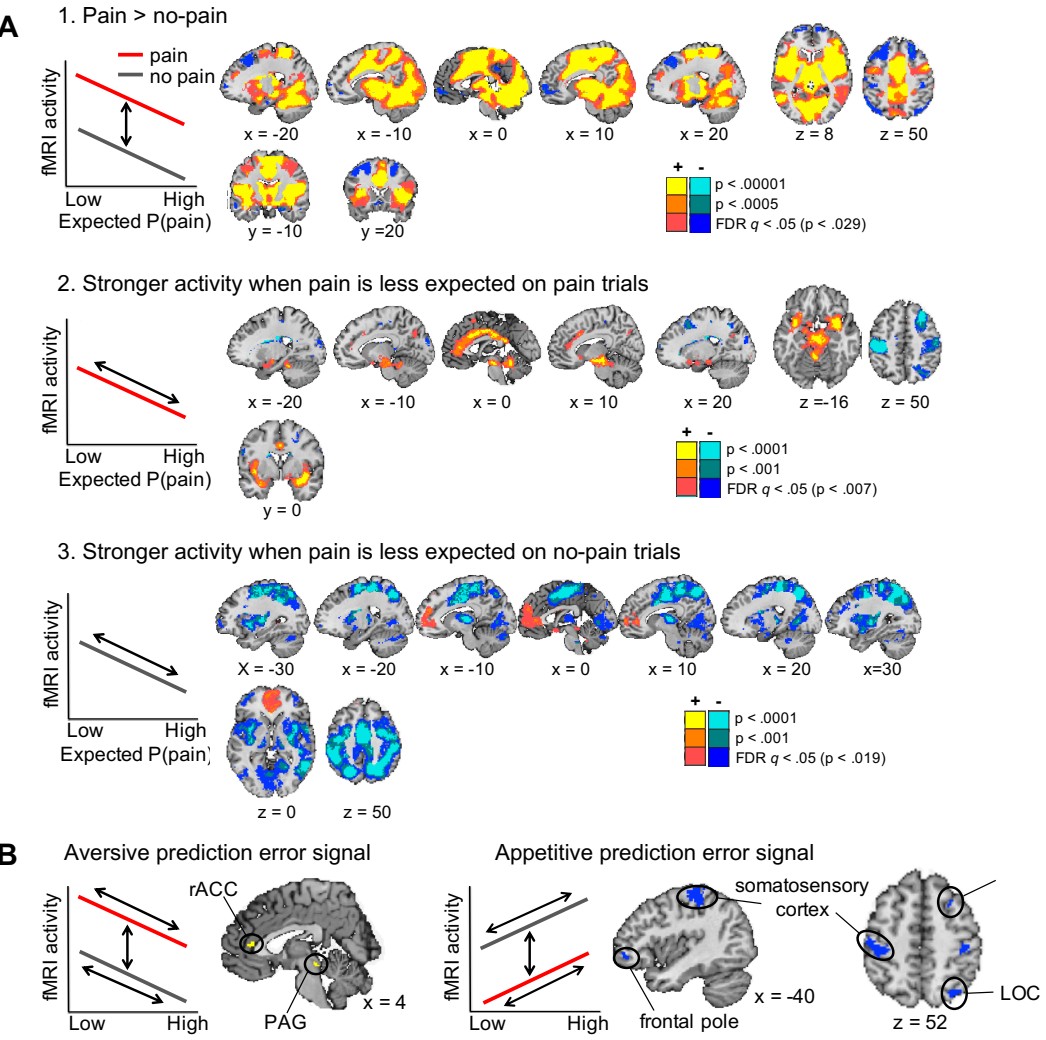

**Appendix 2—figure 1.** Axiomatic tests of brain activation encoding general aversive and appetitive prediction errors (*N* =74). (**A**) Activation associated with the three axioms for aversive prediction errors in our task. Yellow regions showed the effects illustrated in the left panels, and blue regions showed the reverse effects (i.e., the axioms for appetitive prediction errors). All maps were thresholded at *q* < 0.05, false discovery rate (FDR) corrected for multiple comparisons across the whole brain, with higher voxel thresholds superimposed for display. (**B**) Conjunction results. Regions activated for each of the above three contrasts, all thresholded at *q* < 0.05 FDR corrected. Yellow and blue regions showed positive and negative responses for each contrast, respectively, thus encoded general aversive and appetitive prediction errors.

A fourth axiom, that applies to both aversive and appetitive prediction errors, is that activation for received and avoided pain should be equivalent if the outcome is fully predicted (i.e., when the prediction error is zero). As outcomes could never be fully predicted in our task, we could not test this axiom.

## Axiom 1

A large part of the brain fulfilled the first axiom for aversive prediction errors (stronger response to received than avoided pain), including typical pain-processing regions such as the dorsal ACC, (pre)motor cortex, anterior and posterior insula, and thalamus, as well as occipital (visual) cortex (*Appendix 2—figure 1A*, upper panel). We found the opposite effect (stronger response to avoided than received pain) in regions of the ventromedial prefrontal cortex (vmPFC), dorsolateral prefrontal cortex (dlPFC), somatosensory cortex, posterior ACC, and later occipital cortex (LOC).

## Axiom 2

A test of the second axiom for aversive prediction errors (stronger responses to more unexpected pain) revealed several activation clusters, including regions in the vmPFC, dorsal ACC, insula, amygdala, and a midbrain area covering part of the PAG (*Appendix 2—figure 1A*, middle panel). In addition, several other regions, including the right dlPFC and bilateral somatosensory cortex, showed the opposite effect (stronger responses to more expected pain).

## Axiom 3

The third axiom for aversive prediction errors (stronger responses to more expected pain avoidance) was fulfilled by a few regions in the vmPFC and rostral ACC (rACC), as well as part of the PAG. We also found the opposite effect (stronger responses to more unexpected pain avoidance) in several regions, including the dorsal ACC, sensorimotor cortex, thalamus, putamen, and insula.

## Conjunction

A conjunction analysis of the three contrasts reported above (*Appendix 2—figure 1B*) revealed two brain regions that satisfied all three axioms for aversive prediction errors: A midbrain region including part of the PAG (16 voxels) and an area in the rostral ACC (24 voxels). Importantly, we also identified activation that showed a *negative* effect for all three axioms—thus encoding appetitive-like prediction errors—in bilateral somatosensory cortex (433 and 65 voxels in the left and right hemisphere, respectively), left frontopolar cortex (47 voxels), right dlPFC (middle frontal gyrus; 27 voxels), and right LOC (253 voxels).

Together, these results replicate our previous finding that the PAG encodes general aversive prediction errors (*Roy et al., 2014*), and suggest a role for the rostral ACC in encoding aversive prediction errors as well. Furthermore, the identification of an additional neural circuit encoding appetitive-like prediction errors provides an important extension to our previous results, possibly owing to the larger number of participants and hence higher power in the present study.

