## [Editor Report]

This manuscript is of particular interest to readers in the field of pain research. The identification of separate brain systems associated with learning from unexpected pain and learning from unexpected pain relief contributes to understanding of pain avoidance learning. The combination of behavioral and neuroimaging data and computational modeling analyses provide support for the central claims of the paper.

---

## [Decision Letter]

**Decision letter after peer review:**

Thank you for submitting your article "Different brain systems support the aversive and appetitive sides of human pain-avoidance learning" for consideration by *eLife*. Your article has been reviewed by 2 peer reviewers, and the evaluation has been overseen by a Reviewing Editor and Christian Büchel as the Senior Editor. The following individual involved in review of your submission has agreed to reveal their identity: Susanne Becker (Reviewer #1).

Essential revisions:

1) The results indicate that the pharmacological manipulation influenced participants' learning rates, but had no effect on either model-free behavioral indices or neural activity. The authors should discuss potential limitations of the experimental design (e.g., use of a between- versus within- subjects design, relatively small sample size, the absence of a pharmacological manipulation check or analysis of individual differences in drug response) that may have diminished the ability to produce or detect effects, and may want to temper the claims based on these null findings as a result.

2) Effect sizes should be reported for all behavioral analyses.

3) The depiction of the fMRI results should more clearly show the activations described and interpreted within the text. Depiction of corrected results may provide a clearer/more accurate depiction.

4) Given the susceptibility of the BOLD signal within midbrain regions to physiological noise, the authors should describe whether steps were taken during the fMRI preprocessing to remove artifacts that might stem from such noise, and if not, should consider performing such analyses (which can be conducted using approaches that do not require collection of physiological data).

5) The authors need to describe more clearly why the behavioral effects revealed through computational model-based analyses are not evident within model-free behavioral metrics. Relatedly, the description of the computational modeling analyses and the conceptual motivation for the analyses (e.g., what a parameter recovery analysis tells us) should be comprehensible to a reader who is not an expert in computational modeling. This is particularly important as the observed effects of the pharmacological manipulation depend so critically on computational model-based analyses. Additional simulations or use of alternative model-free behavioral metrics could be included in the manuscript to support this explanation. Analysis of choice RT data might also provide additional insights beyond the choice data itself.

*Reviewer #1 (Recommendations for the authors):*

The manuscript targets a research question of high interest, in particular to the pain community. I highly appreciate the combined approach used here of behavioral assessments with pharmacological manipulations, fMRI, and computational modelling. I'm convinced that this combination brings us to a much higher level of potential insights and is an urgently needed approach in the field of pain research. Nevertheless, I have some additional concerns as outlined below.

‒ I have some troubles following the authors' reasoning to assume a general aversive prediction error related to unexpected pain and unexpected pain relief (although I'm aware of the authors previous publication, Roy et al., 2014). Pain relief is usually considered as negative reinforcement and thus to have positive and approach-inducing characteristics. This characteristics are used e.g. in animal models in conditioned place preference and have been also described in human pain testing. Thus, prediction errors related to unexpected pain and prediction errors related to unexpected pain relief like have a different quality. Importantly, when it comes to dopaminergic mechanisms, it has been convincingly described that most dopaminergic populations respond with a pause in spontaneous firing in response to unexpected aversive stimuli and with phasic burst firing in response to unexpected appetitive stimuli. Since these dopaminergic populations seem to be the majority compared to populations responding with burst activity to aversive stimuli, it can be assumed that these will be the responses that can be assessed in human (in vivo) testing. The pain relief as operationalized in this experiment, i.e. the omission of a pain stimuli and not a decrease from ongoing pain, might be perceived as such a appetitive stimulus. Such consideration are important related to this manuscript in (at least) two aspects: (1) the dopaminergic manipulation might have differential effects for unexpected pain and unexpected pain relief. If both prediction errors are analyzed together in terms of a general aversive prediction error, these effects might not be observable. Potential differential effects should be considered and discussed. (2) These thoughts are reflected in the result that the model with two learning rates explains the data better than the model with only one learning rate. While I highly appreciate the replication of previous results, I would suggest to largely shorten the presentation of the results (behavioral and fMRI) on the model with only one learning rate. This would make the manuscript more concise and in my opinion more straight forward.

‒ Overall, I find the description of the rationale on which the hypotheses regarding the pharmacological manipulation are based not sufficient. As described above, related to dopamine different scenarios with in part opposing effects are conceivable. These should be considered and discussed in more detail. With regard to the expected effects of naltrexone, I'm missing a more precise rational why naltrexone is expected to affect learning from unexpected pain and unexpected pain relief. I do not find this reasoning very obvious because endogenous opioids are best know to mediate liking responses. However, such liking not necessarily influences learning, and wanting and liking can be dissociated. The authors refer to results on fear conditioning and fear extinction, but this is very different to the present set-up. In contrast, the known roles of endogenous opioids in pain and pain relief is not discussed.

‒ Also related to the manipulation using naltrexone, the used dose of 50mg is assumed to block the majority of mu-opioid receptors. Opioids, particularly mediated through mu-opioid receptors, are well known to be involved in nociceptive processing with potential strong pain-inhibitory effects. The authors report no effects of the naltrexone on the perception of pain stimuli of certain temperatures before performing the learning task. While similar results have been reported before with healthy volunteers, also often pain inhibitory effects of such a dose of naltrexone has been shown as well. Accordingly, potential effects of naltrexone on pain perception should be acknowledged and discussed.

‒ I highly recommend to improve the depiction of the fMRI results. The chosen way of showing always certain slices (e.g. x=-20, x=-10,.…, z=50) is very systematic and helpful for comparisons between figures. However, some of the activations described in the text are not or only hardly visible, even for regions that appear of high relevance in this context. For example, it cannot be decided based on the figures and the reported results, which parts of the insula (anterior, posterior) are involved (Figure 3). In addition, the choice of showing also uncorrected results in the figures, results in "messy" pictures where in some instances very large parts of the brain appear activated. Further, for the blue shading I'm almost not able to see any differences. On a screen where I can zoom in it is ok, but on a print-out it is indistinguishable. Since in the text, only the thresholded results are reported and discussed, I would prefer also to only see them in the figures.

‒ On result of the conjunction analyses for the three axioms related to the assumed general prediction error, is an activation in the PAG. However, midbrain regions are particularly susceptible to artefacts induced, for example, by physiological noise. If I'm correct the authors did not used any cleaning/correction methods for such potential artefacts including physiological noise (which can be also estimated by established methods without recording physiological parameters). This make the interpretation of activation in the midbrain, including the PAG, very problematic. The same applies for the activation reported for pain specific prediction errors (Figure 4C; page 16).

‒ As mentioned also mentioned by the authors, the effect size (particularly considering the between-subject design) is rather low. Based in this, I would strongly recommend to report for all behavioral outcomes independent of the computational modeling, effect sizes. Do not use partial eta squared as it largely overestimates effects in particular with small sample sizes. A better measure is, for example, generalized eta squared.

‒ Page 7, line 15: without having read the methods section, it is hard to understand what the "hyperparameters" represent. I'd suggest to rephrase this sentence to make the manuscript easily accessible also for readers without a background on computational modeling. Similarly, I'd recommend to rephrase the section "parameter recovery" on page 10 to describe it in less technical terms and make it more accessible (as I think this is an important part of the modelling approach).

‒ Please mention based on with reference or similar the subregions of the ACC were named because there are different systems commonly used.

‒ Page 18, line 1: what is meant by secondary outcomes? Do you mean secondary rewards/reinforcers?

‒ Page 21, lines 28-29: the sentence "…in terms of neural processing, avoiding pain is not comparable to gaining a reward" should be rephrased. This is also not what has been done in the present study. Simple main effects in terms of brain activation in response to receiving pain or receiving pain relief (ie. pain omission) was not analyzed here.

‒ Page 28, prior distributions: The exact shape of the priors can influence the model calculations and the resulting posteriors. Thus, the choice of the prior distributions should be justified.

*Reviewer #2 (Recommendations for the authors):*

Jepma et al. report an interesting manuscript studying how we learn from pain and its avoidance. The authors use an instrumental pain avoidance task where participants are required to choose between two stimuli, one of which is followed by painful thermal stimulation to the leg and the other is not. The probabilities of receiving pain drifted across trials using random walks. The authors combined this with pharmacological manipulation of the dopamine (via oral levodopa) or opioid (via oral naltrexone) systems and also with computational modelling of Q-learning rules and neuroimaging via fMRI. So, this is an ambitious and well conceived manuscript.

There are real strengths here. The manuscript is theoretically motivated, addresses a fundamental question about how we learn, and is generally well executed. The task is well controlled, the modelling choices seem appropriate, the imaging and its analyses are broad but well defended and choices in analysis strategies are well defined. The manuscript is well written. I did enjoy reading the manuscript.

The results have some interest. The modelling and neuroimaging data suggest important dissociations between learning about pain and learning about its absence – the modelling suggests faster learning rates for learning from pain than its avoidance. The imaging suggests that these two forms of learning are associated with different networks, with a known network linked to learning about pain but a novel network linked to learning about avoided pain.

These are worthwhile knowledge gains. The idea that different rate parameters govern learning about events that are present versus those that are absent is an old one. It is built into most error-correcting learning rules since Rescorla-Wagner and it makes sense. However, it was useful to see it supported here. The finding that different networks of brain regions were associated with the learning from pain versus avoided of pain was also interesting. The networks linked to the former made sense based on the literature. The networks linked to the latter were more novel and notably did not include classic 'relief' brain regions.

However, there were also important weaknesses here, at least on my readings.

I struggled as a reader to understand how the modelling actually related to the behavior and imaging. That is, there is a real disconnect in the manuscript for me between what is observed (behavior) what is inferred (modelling as well as it basis for correlations with fMRI data).

There were no differences in behavior reported between the two kinds of trials (learning from received pain versus avoided pain) effects, no effects of the drugs on behavioral performance, and no differential effect on learning from received pain versus avoided pain. I have no problems with reporting null effects, but here the reader is left wondering: if there are no behavioral differences reported, then why does the modelling predict that there should be? How accurate is the model given that it clearly predicts slower learning from avoided than received pain in the controls and faster learning from avoided pain under naltrexone and levodopa compared to control? In other words, what is it about the modelling that yields differences in learning rates between the two behavioral conditions and between the vehicle, levodopa, and naltrexone conditions when the behavioral data shown do not? Of course, it could be that the task was too easy – the modelling may be prescient and perhaps possible learning rate differences would be picked up under more difficult (more cues) and weaker probabilistic conditions. Perhaps there are behavioral data (reaction times?) not reported that do actually show differences in learning rate between learning from received pain versus avoided pain or show differences between the drug conditions?

I may have misunderstood all of this and am happy to be corrected. If not, think this issue needs to be addressed and would need new data that is hopefully already in hand to do convincingly (such as choice reaction times) to show some difference in behavior between learning from received pain versus avoided pain and/or some effects of the pharmacological manipulations on these.

In the absence of the data the manuscript seems to have three parts:

1. A more compelling set of findings reporting imaging differences between learning from received pain versus avoided pain that are interesting because they suggest a novel network of brain regions for the latter compared to the literature.

2. A set of null findings that neither pharmacological manipulation affected behavior or these imaging findings.

3. A less compelling set of findings that link the above to possible underlying differences in learning rate parameters.

The first could be of interest but the latter two need to be strengthened, in my opinion.

---

## [Author Response]

Essential revisions:1) The results indicate that the pharmacological manipulation influenced participants' learning rates, but had no effect on either model-free behavioral indices or neural activity. The authors should discuss potential limitations of the experimental design (e.g., use of a between- versus within- subjects design, relatively small sample size, the absence of a pharmacological manipulation check or analysis of individual differences in drug response) that may have diminished the ability to produce or detect effects, and may want to temper the claims based on these null findings as a result.

In the revised Discussion, we more extensively discuss limitations of our study that may explain why we did not find effects of our pharmacological manipulations on model-free behavioral indices or neural activity, and explicitly state that these null results should be taken with caution.

“It is also possible that our pharmacological manipulations did affect prediction-error related dopamine and/or opioid activity, but that we did not have enough statistical power to detect these effects due to our moderate sample size and between-subject design (24-26 participants per treatment group). […] In addition, future studies could use more than one drug dose to sample the putative inverted Ushape function.”

“Another limitation of our study is the absence of a pharmacological manipulation check, for example via blood samples and/or autonomic or behavioral measures that are known to be influenced by dopamine or opioid activity, such as eye blink rate for dopamine (Jongkees and Colzato 2016). […] Thus, the absence of drug effects on our fMRI results and model-independent performance measures should be taken with caution.”

2) Effect sizes should be reported for all behavioral analyses.

We added effect sizes to the behavioral analyses (p. 6-7).

3) The depiction of the fMRI results should more clearly show the activations described and interpreted within the text. Depiction of corrected results may provide a clearer/more accurate depiction.

We added additional slices to original Figure 3 (which is now shown in Appendix 2—figure 1), to more clearly show the activations described in the text. Note that this figure does not show uncorrected results. Instead, in addition to the FDR-corrected results, we also displayed results at *more conservative* (higher) thresholds. We did this because some of the contrasts shown in this figure (especially pain > no pain) activated very large parts of the brain, and these activations look somewhat less ‘blurry’ at a higher threshold. We mention this in the figure legend (“.., with higher voxel thresholds superimposed for display”). We also changed the shades of blue/green displaying the negative activations, to make these easier to dissociate.

We also added additional slices to original Figure 4 (which is now Figure 3). For this figure, we removed the uncorrected activations and now only show the FDRcorrected results.

4) Given the susceptibility of the BOLD signal within midbrain regions to physiological noise, the authors should describe whether steps were taken during the fMRI preprocessing to remove artifacts that might stem from such noise, and if not, should consider performing such analyses (which can be conducted using approaches that do not require collection of physiological data).

We did not apply physiological noise correction procedures, and discuss this limitation in the revised manuscript (p. 22):

“In addition, we did not collect physiological data during fMRI scanning; hence could not remove potential artefacts related to cardiac and respiratory processes. […] Furthermore, even if physiological changes in respiratory and cardiac cycle did correlate with the timing of our pain outcomes, this would be unlikely to systematically affect our prediction-error signals as these were based on parametric-modulator regressors that were orthogonal to the main outcome-onset regressors.”

5) The authors need to describe more clearly why the behavioral effects revealed through computational model-based analyses are not evident within model-free behavioral metrics. Relatedly, the description of the computational modeling analyses and the conceptual motivation for the analyses (e.g., what a parameter recovery analysis tells us) should be comprehensible to a reader who is not an expert in computational modeling. This is particularly important as the observed effects of the pharmacological manipulation depend so critically on computational model-based analyses. Additional simulations or use of alternative model-free behavioral metrics could be included in the manuscript to support this explanation. Analysis of choice RT data might also provide additional insights beyond the choice data itself.

We agree that this part of the results needed more explanation. In the revised Results section we more clearly describe possible explanations for our finding that the pharmacological manipulations affected model parameters without affecting model-free performance measures. We also make clearer how our

simulation/parameter recovery analysis address this issue, and summarize the main findings from this analysis in a more intuitive way (pp. 11-12):

“Data simulation and parameter recovery

The fact that our pharmacological manipulations increased ᾱ_no-pain_ (negating the learning asymmetry found in the placebo group) and reduced *β¯* (increasing choice stochasticity) seems at odds with the absence of drug effects on model independent performance measures. […] These findings suggest that levodopa and naltrexone really had two computational effects—an increased learning rate for avoided pain, and an increased degree of choice stochasticity—which combination yielded no significant effects on model independent performance measures.”

In addition, to make our modeling approach more comprehensible for readers without a modeling background, we added an explanation of our hierarchical Bayesian modeling approach and the meaning of ‘group-level parameters’, before presenting the modeling results (p. 8):

“We estimated the model parameters using a hierarchical Bayesian approach. This approach assumes that every participant has a different set of model parameters, which are drawn from group-level distributions. In this way, the information in the individual data is aggregated, while still respecting individual differences (Gelman 2014). Each group-level distribution is in turn governed by a group-level mean and a group-level standard deviation (SD) parameter. These group-level parameters were estimated separately for the placebo, levodopa, and naltrexone groups. As we are primarily interested in differences between treatment groups, our primary variables of interest are the parameters governing the means of the group-level distributions, which we denote with overbars (e.g., ᾱ_pain_ refers to the group-level mean of *α*_pain_).”

Finally, we tested for drug effects on choice RT, and found that ‘stay’ choices following no-pain outcomes were faster than choices following pain outcomes. However, there were no drug effects on choice RT. These results are added to the Results section (p. 7):

“Choice reaction times (RTs) were faster when participants stayed with the same choice option following a no-pain outcome (no pain/stay choices; mean = 729 ms) than when participants either stayed or switched following a pain outcome (mean = 778 and 795, respectively; no pain/stay vs. pain/stay: *t*(82) = 5.0, *p* <.001, Cohen’s *d* = 0.54; no pain/stay vs. pain/switch: *t*(82) = 6.9, *p* < 0.001, Cohen’s *d* = 0.76). Pain/stay and pain/switch RTs did not differ from each other (*t*(82) = 1.7, *p* =.10, Cohen’s *d* = 0.18). The faster no pain/stay choices likely reflect that no-pain outcomes usually indicate that participants are on the right track, which makes staying with that option a relatively simple decision. The interpretation of pain outcomes is less straightforward, as these outcomes could either indicate a change in outcome probabilities—in which case a switch to the other option is warranted—or an occasional ‘unlucky’ outcome due to the probabilistic task nature. There was no treatment effect on RT for no pain/stay, pain/stay or pain/switch trials (*F*(2,80) = 0.35, *p* = 0.70, *η*^-^ = 0.01; *F*(2,80) = 0.68, *p* = 0.51, *η*^-^ = 0.02; *F*(2,80) = 0.93, *p* = 0.40, *η*^-^ = 0.02, respectively).”

Reviewer #1 (Recommendations for the authors):The manuscript targets a research question of high interest, in particular to the pain community. I highly appreciate the combined approach used here of behavioral assessments with pharmacological manipulations, fMRI, and computational modelling. I'm convinced that this combination brings us to a much higher level of potential insights and is an urgently needed approach in the field of pain research. Nevertheless, I have some additional concerns as outlined below.‒ I have some troubles following the authors' reasoning to assume a general aversive prediction error related to unexpected pain and unexpected pain relief (although I'm aware of the authors previous publication, Roy et al., 2014). Pain relief is usually considered as negative reinforcement and thus to have positive and approach-inducing characteristics. This characteristics are used e.g. in animal models in conditioned place preference and have been also described in human pain testing. Thus, prediction errors related to unexpected pain and prediction errors related to unexpected pain relief like have a different quality. Importantly, when it comes to dopaminergic mechanisms, it has been convincingly described that most dopaminergic populations respond with a pause in spontaneous firing in response to unexpected aversive stimuli and with phasic burst firing in response to unexpected appetitive stimuli. Since these dopaminergic populations seem to be the majority compared to populations responding with burst activity to aversive stimuli, it can be assumed that these will be the responses that can be assessed in human (in vivo) testing. The pain relief as operationalized in this experiment, i.e. the omission of a pain stimuli and not a decrease from ongoing pain, might be perceived as such a appetitive stimulus. Such consideration are important related to this manuscript in (at least) two aspects: (1) the dopaminergic manipulation might have differential effects for unexpected pain and unexpected pain relief. If both prediction errors are analyzed together in terms of a general aversive prediction error, these effects might not be observable. Potential differential effects should be considered and discussed. (2) These thoughts are reflected in the result that the model with two learning rates explains the data better than the model with only one learning rate. While I highly appreciate the replication of previous results, I would suggest to largely shorten the presentation of the results (behavioral and fMRI) on the model with only one learning rate. This would make the manuscript more concise and in my opinion more straight forward.

Thanks for these important comments. To address these, we made two changes to the manuscript.

First, we agree that our analyses on general (outcome-nonspecific) aversive prediction error signals is not clearly linked to our dopamine manipulation nor (more generally) to the main questions addressed in this paper. To make the manuscript more concise and straightforward, we moved the corresponding sections and figure (which mostly served as a replication and extension of our previous results) to the appendix. We refer to this analysis on p. 12:

“We also performed an axiomatic analysis to identify brain activation encoding *general* aversive prediction errors […] As this analysis is not directly linked to the research questions addressed in the present study, we report it in appendix 2.”

Second, we extended the dopamine-related hypotheses in the Introduction. We now not only consider the possibility that dopamine bursts mediate learning from unexpected pain absence, but also that dopamine dips mediate learning from unexpected pain (p. 4, 5):

“Specifically, unexpected rewards trigger a phasic increase (burst) in dopamine activity, while the unexpected absence of reward triggers a phasic decrease (dip) in dopamine activity. […] Thus, if phasic dopamine dips support learning from unexpected pain, we expect levodopa to suppress learning rates and neural prediction error signaling when pain is received.”

‒ Overall, I find the description of the rationale on which the hypotheses regarding the pharmacological manipulation are based not sufficient. As described above, related to dopamine different scenarios with in part opposing effects are conceivable. These should be considered and discussed in more detail. With regard to the expected effects of naltrexone, I'm missing a more precise rational why naltrexone is expected to affect learning from unexpected pain and unexpected pain relief. I do not find this reasoning very obvious because endogenous opioids are best know to mediate liking responses. However, such liking not necessarily influences learning, and wanting and liking can be dissociated. The authors refer to results on fear conditioning and fear extinction, but this is very different to the present set-up. In contrast, the known roles of endogenous opioids in pain and pain relief is not discussed.

We extended the hypotheses regarding the effects of levodopa, as described in (the second part of) our reply to the previous comment*.*

With regard to the expected effects of naltrexone, I'm missing a more precise rational why naltrexone is expected to affect learning from unexpected pain and unexpected pain relief. I do not find this reasoning very obvious because endogenous opioids are best know to mediate liking responses. However, such liking not necessarily influences learning, and wanting and liking can be dissociated. The authors refer to results on fear conditioning and fear extinction, but this is very different to the present set-up. In contrast, the known roles of endogenous opioids in pain and pain relief is not discussed.

We did not discuss the well-known roles of endogenous opioids in pain relief and liking responses because it is unknown whether these processes influence learning (the topic of our paper). However, in the revised Introduction, we mention these roles of endogenous opioids, and suggest that these could potentially affect pain-avoidance learning as well (p. 5):

“In humans, endogenous opioids are well-known to mediate pain relief (Levine et al. 1978, Eippert et al. 2009) as well as pleasure and liking responses (Berridge and Kringelbach 2015, Nummenmaa and Tuominen 2018), but it is currently unknown whether the roles of the opioid system in affective valuation also influence painrelated *learning*. One interesting possibility—that remains to be explored—is that endogenous opioids support pain-avoidance learning by enhancing the pleasant feeling of relief following successful pain avoidance (Sirucek et al. 2021).”

‒ Also related to the manipulation using naltrexone, the used dose of 50mg is assumed to block the majority of mu-opioid receptors. Opioids, particularly mediated through mu-opioid receptors, are well known to be involved in nociceptive processing with potential strong pain-inhibitory effects. The authors report no effects of the naltrexone on the perception of pain stimuli of certain temperatures before performing the learning task. While similar results have been reported before with healthy volunteers, also often pain inhibitory effects of such a dose of naltrexone has been shown as well. Accordingly, potential effects of naltrexone on pain perception should be acknowledged and discussed.

We indeed found no effects of naltrexone on perceived heat pain immediately prior to the learning task. However, we agree that potential effects of naltrexone on pain perception during the learning task cannot be excluded, and acknowledge and discuss this in the revised Discussion (p. 21):

“A limitation of our experimental design is that we did not acquire pain ratings during the pain-avoidance learning task. […] However, we believe that it is unlikely that naltrexone affected participants’ pain sensitivity because (i) we found no effects of naltrexone on pain ratings immediately prior to the learning task, and (ii) previous studies suggest that opioid antagonists rarely affect pain perception in experimental pain paradigms (Grevert and Goldstein 1977, Eippert et al. 2008, Werner et al. 2015, Sirucek et al. 2021).”

‒ I highly recommend to improve the depiction of the fMRI results. The chosen way of showing always certain slices (e.g. x=-20, x=-10,.…, z=50) is very systematic and helpful for comparisons between figures. However, some of the activations described in the text are not or only hardly visible, even for regions that appear of high relevance in this context. For example, it cannot be decided based on the figures and the reported results, which parts of the insula (anterior, posterior) are involved (Figure 3). In addition, the choice of showing also uncorrected results in the figures, results in "messy" pictures where in some instances very large parts of the brain appear activated. Further, for the blue shading I'm almost not able to see any differences. On a screen where I can zoom in it is ok, but on a print-out it is indistinguishable. Since in the text, only the thresholded results are reported and discussed, I would prefer also to only see them in the figures.

We added additional slices to the figures showing our fMRI results, no longer show uncorrected results, and changed the blue shading. See our reply to Essential Revision 3 for more details.

‒ On result of the conjunction analyses for the three axioms related to the assumed general prediction error, is an activation in the PAG. However, midbrain regions are particularly susceptible to artefacts induced, for example, by physiological noise. If I'm correct the authors did not used any cleaning/correction methods for such potential artefacts including physiological noise (which can be also estimated by established methods without recording physiological parameters). This make the interpretation of activation in the midbrain, including the PAG, very problematic. The same applies for the activation reported for pain specific prediction errors (Figure 4C; page 16).

We acknowledge and discuss that we did not apply physiological noise correction procedures in the revised Discussion (p. 22). Also see our reply to Essential Revision 4.

‒ As mentioned also mentioned by the authors, the effect size (particularly considering the between-subject design) is rather low. Based in this, I would strongly recommend to report for all behavioral outcomes independent of the computational modeling, effect sizes. Do not use partial eta squared as it largely overestimates effects in particular with small sample sizes. A better measure is, for example, generalized eta squared.

We added effect sizes to the behavioral analyses (p. 6-7). For the one-way ANOVA’s we report eta squared, which equals generalized eta squared (and partial eta squared) in case of one-way ANOVA’s. For t-tests we report Cohen’s *d*.

‒ Page 7, line 15: without having read the methods section, it is hard to understand what the "hyperparameters" represent. I'd suggest to rephrase this sentence to make the manuscript easily accessible also for readers without a background on computational modeling. Similarly, I'd recommend to rephrase the section "parameter recovery" on page 10 to describe it in less technical terms and make it more accessible (as I think this is an important part of the modelling approach).

In the revised manuscript we added a more intuitive explanation of our hierarchical Bayesian modeling approach and the meaning of the group-level parameters, earlier on (in the Results section, p. 8). Note that we no longer use the more technical term ‘hyperparameters’ here.

“We estimated the model parameters using a hierarchical Bayesian approach. […] As we are primarily interested in differences between treatment groups, our primary variables of interest are the parameters governing the means of the group-level distributions, which we denote with overbars (e.g., ᾱ_pain_ refers to the group-level mean of *α*_pain_).”

We also rewrote the parameter-recovery section to make it more accessible for readers without a modeling background. See our reply to Essential Revision 5 for the revised section.

‒ Please mention based on with reference or similar the subregions of the ACC were named because there are different systems commonly used.

Our use of the terms rostral and dorsal ACC is based on, for example, Bush,

Luu and Posner (2000, TiCS). We are also aware of the four-region model of the cingulate cortex (e.g., Palomero-Gallagher et al., 2009, HBM)—which uses the terms ACC and midcingulate cortex rather than rostral and dorsal ACC—but decided to use the rostral vs. dorsal ACC terminology as this is common in neuroimaging studies. We mention this on p. 12 of the revised manuscript:

“When referring to subdivisions of the anterior cingulate cortex (ACC), we use the terms rostral and dorsal ACC, as is common in neuroimaging studies (Bush et al. 2000). Note that these two regions have also been referred to as ACC and midcingulate cortex, respectively (Palomero-Gallagher et al. 2009).”

‒ Page 18, line 1: what is meant by secondary outcomes? Do you mean secondary rewards/reinforcers?

We indeed meant secondary reinforcers, and have clarified this in the revised manuscript.

‒ Page 21, lines 28-29: the sentence "…in terms of neural processing, avoiding pain is not comparable to gaining a reward" should be rephrased. This is also not what has been done in the present study. Simple main effects in terms of brain activation in response to receiving pain or receiving pain relief (ie. pain omission) was not analyzed here.

Thanks for pointing this out. We indeed referred to prediction errors (teaching signals), and not to pain or no-pain outcomes per se. We rephrased this sentence to “… in terms of neural processing, learning from avoided pain is not comparable to learning from rewards.”

‒ Page 28, prior distributions: The exact shape of the priors can influence the model calculations and the resulting posteriors. Thus, the choice of the prior distributions should be justified.

Our choice of priors was based on Ahn et al. (2017). We used weakly informative priors as implemented in the hBayesDM package, and added this to the ‘Prior distributions’ section. We also added more references to this section (Wetzels et al. 2010, Ahn et al. 2014, Ahn et al., 2017).

Reviewer #2 (Recommendations for the authors):[…]However, there were also important weaknesses here, at least on my readings.I struggled as a reader to understand how the modelling actually related to the behavior and imaging. That is, there is a real disconnect in the manuscript for me between what is observed (behavior) what is inferred (modelling as well as it basis for correlations with fMRI data).

We address this comment in our replies below.

There were no differences in behavior reported between the two kinds of trials (learning from received pain versus avoided pain) effects, no effects of the drugs on behavioral performance, and no differential effect on learning from received pain versus avoided pain. I have no problems with reporting null effects, but here the reader is left wondering: if there are no behavioral differences reported, then why does the modelling predict that there should be? How accurate is the model given that it clearly predicts slower learning from avoided than received pain in the controls and faster learning from avoided pain under naltrexone and levodopa compared to control? In other words, what is it about the modelling that yields differences in learning rates between the two behavioral conditions and between the vehicle, levodopa, and naltrexone conditions when the behavioral data shown do not? Of course, it could be that the task was too easy – the modelling may be prescient and perhaps possible learning rate differences would be picked up under more difficult (more cues) and weaker probabilistic conditions. Perhaps there are behavioral data (reaction times?) not reported that do actually show differences in learning rate between learning from received pain versus avoided pain or show differences between the drug conditions?I may have misunderstood all of this and am happy to be corrected. If not, think this issue needs to be addressed and would need new data that is hopefully already in hand to do convincingly (such as choice reaction times) to show some difference in behavior between learning from received pain versus avoided pain and/or some effects of the pharmacological manipulations on these.

First, we would like to clarify that there were differences in behavior between trials on which pain was received versus avoided: (i) trials on which pain was received were more often followed by a switch to the other choice option, in all treatment groups (p. 6); (ii) trials on which pain was received were associated with slower subsequent choice RTs, in all treatment groups (p. 7, see the last part of this reply): and (iii) model-estimated learning rates in the placebo group were higher when pain was received than avoided (p. 8 and Figure 2A).

That said, the Reviewer is right that we found no drug effects on model-independent performance measures, which appears to be at odds with the drug effects on model parameters. We agree that this is puzzling, and this also made us wonder whether the drug effects on the model parameters may have been due to an artefact in our modeling procedure. We addressed this issue by means of simulation and parameter-recovery analyses, and explain the rationale for and results of this analyses more clearly on p. 11-12 of the revised manuscript (see our reply to Essential Revision 5). In sum, these analyses suggest that the drugs affected both the learning process and the degree of choice stochasticity, and that these two computational effects cancelled each other out such that their combination resulted in no net effect on model-independent performance measures. Thus, computational modeling might have picked up more subtle behavioral differences that are difficult to observe with a “naked eye” (e.g., Huys et al., 2021, Neuropsychopharmacology). However, we also acknowledge the possibility that our null results regarding the pharmacological manipulations were due to limitations of our experimental design (relatively small sample size, use of a between-subjects design, no analysis of individual differences in drug response). We discuss these limitations in the revised Discussion, and state that the absence of drug effects have to be taken with caution (p. 20, 21; also see our reply to Essential Revision 1).

Finally, we thank the Reviewer for the suggestion to analyze reaction times.

We did this, and report the results on p. 7 (see our last reply to Essential Revision 5).

In the absence of the data the manuscript seems to have three parts:1. A more compelling set of findings reporting imaging differences between learning from received pain versus avoided pain that are interesting because they suggest a novel network of brain regions for the latter compared to the literature.2. A set of null findings that neither pharmacological manipulation affected behavior or these imaging findings.3. A less compelling set of findings that link the above to possible underlying differences in learning rate parameters.The first could be of interest but the latter two need to be strengthened, in my opinion.

We agree that the lack of drug effects on model-independent behavioral measures and fMRI activity are the less compelling part of this paper. Therefore, we have focused this paper on our most compelling set of findings that is indicative of two separate systems for pain-avoidance learning. However, we feel that it is important to report the less compelling part as well, as this can be informative for future pharmacological studies. As described above, we also discuss various possible explanations for the absence of drug effects on fMRI and model independent behavioral measures, and state that these null effects have to be taken with caution.